# Integrative spatial and genomic analysis of tumor heterogeneity with Tumoroscope

Shadi Shafighi[1,2,3], Agnieszka Geras[1,4,5,6], Barbara Jurzysta[1], Alireza Sahaf Naeini[1], Igor Filipiuk[1], Alicja Rączkowska[1], Hosein Toosi[7], Łukasz Koperski[8], Kim Thrane[9], Camilla Engblom [10,11], Jeff E. Mold[10], Xinsong Chen [12], Johan Hartman [12,13], Dominika Nowis [14], Alessandra Carbone [2,15], Jens Lagergren [7,17] & Ewa Szczurek [1,16,17] ✉

Spatial and genomic heterogeneity of tumors are crucial factors influencing cancer progression, treatment, and survival. However, a technology for direct mapping the clones in the tumor tissue based on somatic point mutations is lacking. Here, we propose Tumoroscope, the first probabilistic model that accurately infers cancer clones and their localization in close to single-cell resolution by integrating pathological images, whole exome sequencing, and spatial transcriptomics data. In contrast to previous methods, Tumoroscope explicitly addresses the problem of deconvoluting the proportions of clones in spatial transcriptomics spots. Applied to a reference prostate cancer dataset and a newly generated breast cancer dataset, Tumoroscope reveals spatial patterns of clone colocalization and mutual exclusion in sub-areas of the tumor tissue. We further infer clone-specific gene expression levels and the most highly expressed genes for each clone. In summary, Tumoroscope enables an integrated study of the spatial, genomic, and phenotypic organization of tumors.

Tumor evolution proceeds by the accumulation of mutations, resulting in the emergence of distinct cancer cell subpopulations, called *clones*, characterized by their genotypes. The spatial distribution of these clones may vary drastically across tumor tissue. This *genetic* and *spatial* tumor heterogeneity are the two key determinants of patient prognosis, survival, and treatment[1–3]. Characterization of the *phenotypic* heterogeneity of tumors, i.e., linking the potential differences

between expression profiles of clones and their spatial distribution, remains largely unexplored.

Most research on intra-tumor heterogeneity relies on bulk DNA sequencing (DNA-seq) or single-cell DNA-seq (scDNA-seq) data[4,5]. However, bulk DNA-seq measures a mixture of millions of cells from a given sample, which may contain both cancer and healthy cells, yielding only aggregated variant allele frequency information. Several

[1]Faculty of Mathematics, Informatics and Mechanics, University of Warsaw, Warsaw, Poland. [2]Sorbonne Universite, CNRS, IBPS, Laboratoire de Biologie Computationnelle et Quantitative, Paris, France. [3]Cancer Research UK Cambridge Institute, Cambridge, UK. [4]Faculty of Mathematics and Information Science, Warsaw University of Technology, Warsaw, Poland. [5]Department of Statistics, Columbia University, New York, NY 10027, USA. [6]Irving Institute for Cancer Dynamics, Columbia University, New York, NY 10027, USA. [7]SciLifeLab, School of EECS, KTH Royal Institute of Technology, Stockholm, Sweden. [8]Department of Pathology, Medical University of Warsaw, Warsaw, Poland. [9]Department of Gene Technology, KTH Royal Institute of Technology, SciLifeLab, Stockholm, Sweden. [10]Department of Cell and Molecular Biology, Karolinska Institutet, Solna, Sweden. [11]SciLifeLab, Department of Medicine Solna, Center of Molecular Medicine, Karolinska Institute and University Hospital, Stockholm, Sweden. [12]Department of Oncology-Pathology, Karolinska Institutet, Stockholm, Sweden. [13]Department of Clinical Pathology and Cancer Diagnostics, Karolinska University Hospital, Stockholm, Sweden. [14]Laboratory of Experimental Medicine, Medical University of Warsaw, Warsaw, Poland. [15]Institut Universitaire de France, Paris, France. [16]Institute of AI for Health, Helmholtz Munich, German Research Center for Environmental Health, Neuherberg, Germany. [17]These authors contributed equally: Jens Lagergren, Ewa Szczurek. ✉e-mail: szczurek@mimuw.edu.pl

methods exist for clonal deconvolution of bulk DNA-seq data, aiming to reconstruct clone genotypes, clone frequencies, and their phylogenetic relationships[6–11]. Recently, techniques for identifying clonal evolution using mutations found in scDNA-seq[12] or a combination of bulk and single cell RNA sequencing (scRNA-seq)[13–15] have emerged. Despite technological progress[16], scDNA-seq remains more labor-intensive, less accurate, and costlier than the well-established bulk DNA-seq[17]. A significant drawback of both bulk and scDNA-seq is the requirement for tissue disaggregation, resulting in the loss of spatial information. As DNA-seq-based methods, they are incapable of elucidating phenotypic heterogeneity.

The localization of cancer clones has been previously investigated using multi-region single-cell or bulk DNA-seq, combined with computational inference of clonal populations for each region[4,18–20]. However, this method is inherently coarse-grained, as each region represents a bulk sample, consisting of multiple clones with unresolved spatial positioning within the tissue. Currently, no experimental technique exists for large-scale sequencing of single-cell DNA in situ. Still, the advent of spatial transcriptomics (ST) enables spatially resolved RNA-seq from small groups of 1–100 cells, localized within spots on an ST array[21,22]. The number of cells per spot can vary due to differences in cell size, density, and specific ST technology parameters, such as spot diameter. ST thus allows for the investigation of spatial gene expression patterns across tissues. Although the resolution of ST is considerably higher than multi-region bulk sequencing, it still only provides aggregated signals from cell mixtures. Recent methods have been developed to map clonal copy number alterations using ST data; however, these approaches do not account for somatic point mutations, which are key drivers of tumor evolution[23,24]. For certain cancers, minimal or no copy number alterations occur during disease progression, rendering these methods ineffective for detecting critical clones driven by somatic point mutations[25]. Since ST is an RNA-seq-based protocol without single-cell resolution, inferring the genotypes of point mutations within clones at each spot is challenging.

Finally, the phenotypes of individual tumor cells are typically studied using scRNA-seq. However, this technology does not measure the DNA of these cells, leaving phenotypic data unlinked to the corresponding cancer clones. In conclusion, no current approach effectively integrates tumor genetic, spatial, and phenotypic heterogeneity at near-single-cell resolution.

To tackle this challenge, we introduce Tumoroscope, a probabilistic graphical model that utilizes somatic point mutation data from ST reads, clone genotypes reconstructed from bulk DNA-seq, and cancer cell counts in the spots annotated in hematoxylin and eosin-stained (H&E) images to unravel the clonal composition of each spot within the tumor sample. This approach enables us to spatially locate somatic point mutations and clones derived from DNA-seq within the tissue. Additionally, we develop a regression model to infer gene expression profiles of the clones. Following validation of Tumoroscope using simulated data, we analyze newly generated breast cancer dataset and a previously published prostate cancer dataset[26] to address crucial questions regarding co-localization and mutual exclusion patterns in the spatial arrangement of clones and their phenotypes.

## Results

Tumoroscope is a probabilistic framework designed to map cancer clones across tumor tissues by integrating signals from H&E stained images (Fig. 1a), bulk DNA-seq (Fig. 1b), and spatially-resolved transcriptomics (Fig. 1c). The data preprocessing pipeline begins with a two-step analysis of the H&E-stained tissue image (Fig. 1d). Initially, ST spots situated within cancer cell-containing regions are identified. Subsequently, we estimate the number of cells present in each of these ST spots (using custom QuPath[27] scripts; Methods). We then proceed to reconstruct cancer clones, including their frequencies and genotypes. This is accomplished using somatic mutations and allele-specific copy number data derived from bulk DNA-seq data (utilizing existing methods: Vardict[28], FalconX[29], and Canopy[9], see Methods, Fig. 1e).

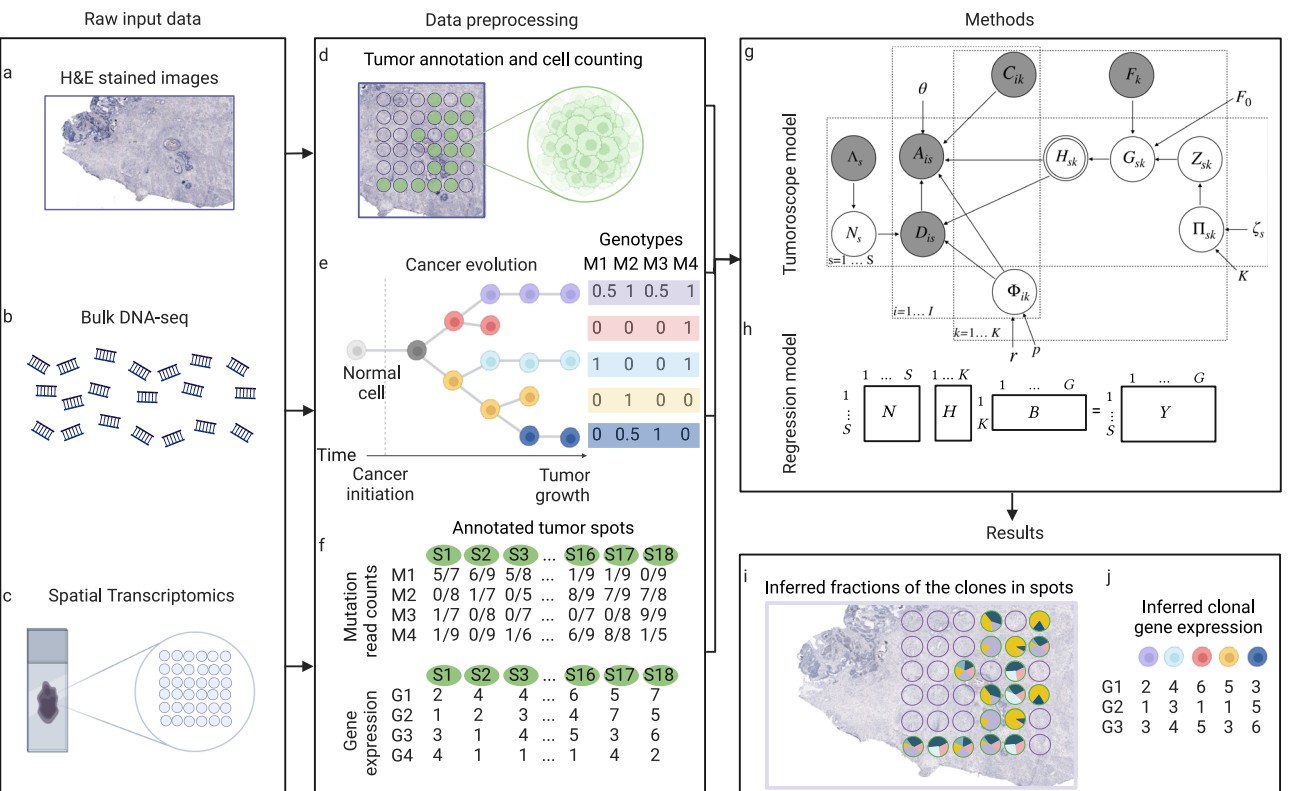

**Fig. 1 | Overview of the Tumoroscope framework. a–c** Input data. **d-f** Data preprocessing. **g** Tumoroscope probabilistic model. **h** Regression model for inferring gene expression profiles of the clones. **i** Results of Tumoroscope. **j** Output of the regression model. Figure is created in BioRender[54].

The binary values of these genotypes are scaled by the ratio of the major copy number to the total copy number, yielding values between 0 and 1. Next, we analyze the data in terms of the number of alternated reads and the total number of reads (mutation coverage), along with gene expression observed in each tumor-indicated spot (Fig. 1f). As we use the Binomial distribution for read counts and model the expected ratio of alternative reads to total reads, our model remains robust against gene expression fluctuations. Tumoroscope's core assumption is that each ST spot contains a hidden mixture of clones reconstructed from bulk DNA-seq data. The method utilizes: i) estimated cell counts per spot as priors, ii) alternated and total read counts for mutations in ST spots, and iii) clone genotypes and frequencies through a probabilistic deconvolution model (Fig. 1g). Tumoroscope's output identifies the proportions of clones in each spot (Fig. 1i). Additionally, it refines the prior cell counts estimated from H&E images for each spot using ST data inference. Lastly, we employ a regression model with gene expression data as independent variables and inferred clone proportions in ST spots as dependent variables (Fig. 1h) to deduce gene expression profiles of the clonal populations (Fig. 1j).

## Tumoroscope accurately estimates clone proportions in each spot and demonstrates robustness to input cell count noise

To assess Tumoroscope's performance with known ground truth, we evaluated its accuracy in estimating clone proportions within spots using simulated data. We varied simulation parameters including the number of mutations in clones, the expected number of clones per spot, and the average spot coverage (defined as the total read count summed over all variants per spot, averaged across all spots). We began with a basic setup featuring five clones in the evolutionary tree, 30 mutations in the genotype matrix, an average of 13.6 mutations per clone, and an expected 2.5 clones per spot. We then created four additional setups by adjusting the average mutations per clone to 5.1 and 15, and the expected clones per spot to 1 and 4.5, respectively. To examine the impact of average spot coverage, we further varied the spot coverage for each of these five simulation setups, establishing very low, low, medium, and high coverage levels, corresponding to average read counts per spot of 18, 50, 80, and 110, respectively (Methods). We generated 10 datasets for each of the 20 setups resulting from the five aforementioned configurations and four coverage levels, totaling 200 distinct simulated datasets for evaluation (see Supplementary Table 1 for detailed simulation setup specifications). To test our model's robustness to noise in cell counts per spot, we considered three noise levels in this input and two Tumoroscope versions, differing by how this input is processed. The model either received true simulated cell numbers per spot as input (corresponding to zero noise level) or we introduced small and large additive noise to these numbers (Methods). In the default version, referred to as Tumoroscope, provided cell counts were used as priors, and the number of cells per spot was inferred considering all available data. In the simplified version, Tumoroscope-fixed, cell number values were provided to the model as fixed input. Both model variations were evaluated for the three noise levels on each of the 200 simulated datasets, resulting in inference for 1200 synthetic datasets in total (Fig. 2). Performance was assessed by calculating the Mean Average

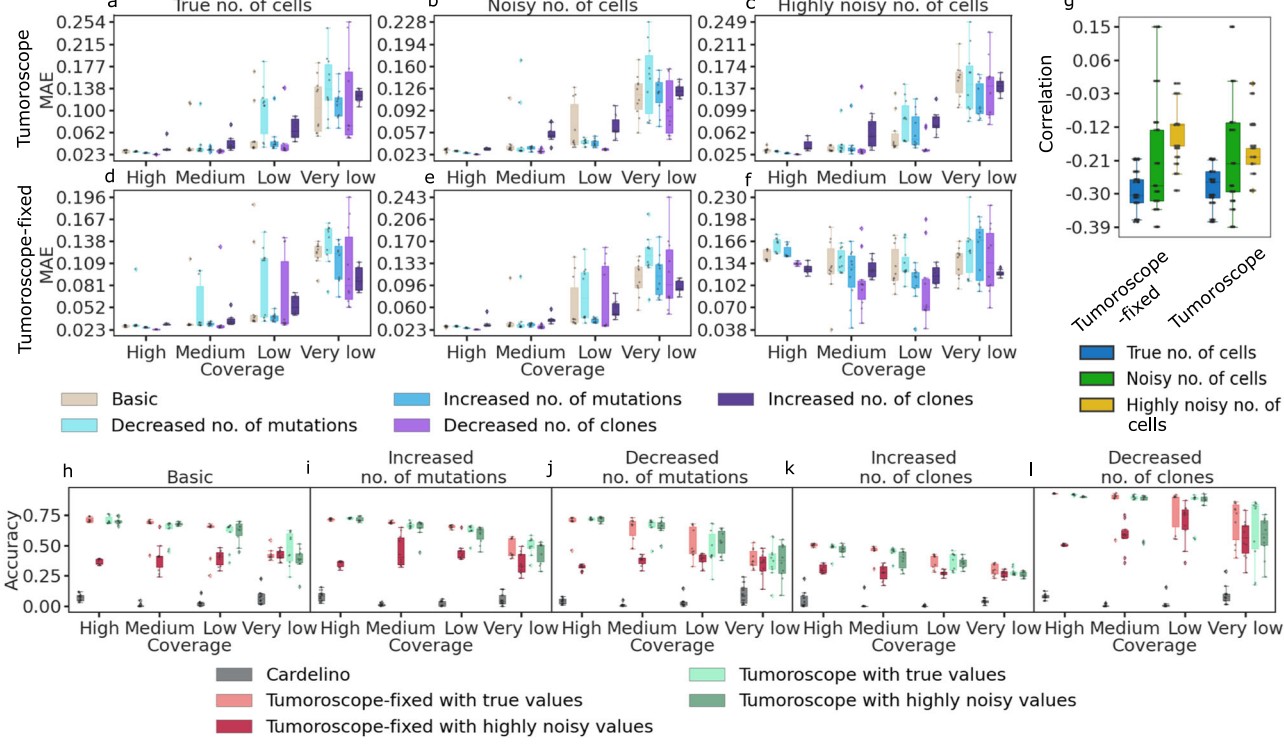

**Fig. 2 | Performance of Tumoroscope on simulated data featuring 5 clones and 30 mutations. a–c** Mean Average Error (MAE; y-axis) as a function of spot coverage (x-axis) in different simulation setups (colors) for Tumoroscope, for different noise levels in the cell count provided at input: no noise (**a**), small noise (**b**) and high noise (**c**). **d-f** The same as in **a–c**, but for Tumoroscope-fixed. **g** Pearson correlation (y-axis) between the average spot coverage and the average error in all the setups is negative for both model versions (x-axis), regardless of the noise in the number of cells provided as input (colors). **h-l** Comparison of the accuracy (y-axis) of the model between cardelino (gray) and two versions of the model given true and highly noisy values for the number of cells (colors), depending on the spot coverage (x-axis), in different simulation setups: basic (**h**), increased (**i**) and decreased (**j**) number of mutations, increased (**k**) and decreased (**l**) number of clones. In each panel, the lower and upper boundaries of the box represent the first (Q1) and third quartiles (Q3), with the median indicated by a line inside the box. The whiskers typically extend to the most extreme data points within 1.5 times the interquartile range (IQR) from the quartiles. Data points outside this range are considered outliers and are plotted individually by diamonds. The boxplots in panels (**a-f**) and (**h-l**) are based on 10 data points each, corresponding to 10 generated datasets for each setup. In panel **g**, each boxplot represents 20 data points, corresponding to the Pearson correlations calculated across the 10 datasets for the 20 different setups.

Error (MAE), representing the average difference between inferred proportions of all clones and their true values for each spot. This process was repeated for all spots, with resulting values averaged across spots.

For both model versions, Tumoroscope and Tumoroscope-fixed, the error increased as the spot coverage decreased (Fig. 2a-f). This trend suggests that deeper sequencing of spots in ST data leads to more accurate clone deconvolution. To further validate this relationship, we calculated the Pearson correlation between the average spot coverage and the average error across the 20 setups derived from the five initial setups and four different spot coverage levels. This correlation analysis was performed for both model versions, across three noise levels, and for varying numbers of cells. The consistent negative correlation across all 20 setups confirms a association between read depth and model accuracy. This observation aligns with theoretical expectations, where the probability of observing at least one read increases as the number of reads per cell rises (Supplementary Fig. 1). Additionally, we observed that higher noise levels weakened this correlation, which is expected. Importantly, this association between read depth and accuracy was similar for both Tumoroscope and Tumoroscope-fixed (Fig. 2g).

Tumoroscope obtained low error (median MAE between 0.02 and 0.15, depending on the spot coverage), regardless of the level of noise in the input cell counts per spot (Fig. 2a-c). Notably, in the case when the true simulated cell counts were given as input, Tumoroscope performed equally well as Tumoroscope-fixed, despite the advantage that the latter was given (Fig.2a vs. d). This advantage turned into a bias when the input cell numbers became noisy, and Tumoroscope-fixed obtained a larger MAE than Tumoroscope (Fig. 2b,e,c,f). Similar results were obtained when higher spot coverage was considered (Supplementary Fig. 2). These results highlight the importance of using input cell counts per spot as priors rather than fixing them, particularly when there is noise in the input, as is often the case with real data. Indeed, in practical scenarios, these input cell counts per spot are estimated from H&E images using nuclei detection algorithms. This task becomes particularly challenging when the cells are densely packed and the nuclei overlap (Methods).

Since Tumoroscope relies on matching proportions of spots to clones by single nucleotide variants, non-expressed mutant alleles decrease the statistical signal in the data. ST data does not capture all mutations as only part of the gene is actually sequenced. To investigate the effect of missing mutations on Tumoroscope's performance, we additionally simulated scenarios with 0%, 25%, 50%, and 75% of missing mutations. These simulations confirmed our model's accuracy also with missing data, indicating that it is sufficient when only a subset of mutations is available. (Supplementary Fig. 3).

## Accounting for clonal mixture within ST spots is essential for model performance

To evaluate the importance of considering clonal heterogeneity within individual spots, we conducted a comparative analysis between Tumoroscope and cardelino, a methodology initially developed for clone assignment in single-cell analysis[14]. We applied cardelino by treating each ST spot as an individual cell. While cardelino was not specifically designed for ST data, its clonal assignment efficiency for spots would theoretically match its single-cell performance if spots exhibited clonal homogeneity. This comparative exercise enables us to quantify the detrimental effects of incorrectly assuming homogeneous spot composition.

For the comparative analysis, we identified the predominant clone in each spot (defined as the clone with the highest proportional representation) as determined by Tumoroscope. Accuracy was quantified as the percentage of concordance between the predicted predominant clone and the actual predominant clone in our simulated dataset. Both Tumoroscope and Tumoroscope-fixed variants were evaluated.

Tumoroscope demonstrated the worst-case median accuracy of 0.27 in the simulation scenarios with an increased number of clones and very low read counts, while achieving the best-case accuracy of approximately 0.92 in scenarios with a decreased number of clones and very high read counts. These results substantially exceeded cardelino's performance, which exhibited median accuracy values spanning 0 to 0.09 across for all simulation scenarios (Fig. 2h-l). Consistent with previous observations (Fig. 2a-g), Tumoroscope's accuracy exhibited an inverse relationship with decreasing spot coverage.

Notably, Tumoroscope achieved optimal accuracy in simulation scenarios with decreased number of clones and lowest accuracy with increased number of clones, suggesting that per-spot clone quantity represents a crucial variable determining its performance. While Tumoroscope-fixed demonstrated lower accuracy compared to the default Tumoroscope implementation, particularly when processing highly noisy input cell counts, it nevertheless significantly outperformed cardelino. These findings emphasize the fundamental importance of accounting for clonal heterogeneity within ST spots.

## Tumoroscope deconvolutes spatial clonal composition in breast sample and reveals clone-specific spatial patterns within distinct sub-areas

To elucidate the spatial organization of clonal populations within tumor architecture, we implemented Tumoroscope on a novel dataset comprising three distinct breast carcinoma sections derived from a single patient (Fig. 3a-f). The experimental design incorporated paired deep whole-exome sequencing (WES) and ST analyses (10x Genomics platform) of adjacent tissue layers for each section (Methods). The ST analysis encompassed 4885-4992 spots per sample. During preliminary data processing, we implemented neoplastic spot selection based on expert histopathological evaluation (Fig. 3e) and quantified cell numbers per spot through computational analysis of H&E images. We identified 608 high-confidence somatic single-nucleotide variants (SNVs) through WES analysis that were co-observed in the ST data reads (Methods) and constructed a phylogenetic tree for those SNVs (Methods). This analysis revealed seven distinct clonal populations, including a base clone devoid of somatic mutations (Figs. 4, 3a,b; Supplementary Fig. 4).

We first investigated the obtained evolutionary tree to identify the driver genes harboring mutations within each clone along the branches of the tree. We cross-referenced these genes with the COSMIC dataset, identifying oncogenes, tumor suppressor, and fusion genes, and specifically focusing on gene categories associated with breast cancer, including 'Known hallmark of breast cancer', 'Known breast cancer genes', 'Known mutated genes in cancer', and the 'Top 20 mutated genes in breast cancer' (Fig. 4). In total, our analysis unveiled 48 mutations that manifested in at least one of the genes within these specified categories. Genes *NF1*, *RBM10X*, *RECQL*, and *ERBB2* appeared in multiple categories. For example, *NF1* was recognized as both a tumor suppressor and fusion gene and was present within the 'Known hallmark of breast cancer' and 'Top 20 mutated genes in breast cancer' categories. We detected two distinct mutations of *NF1*, one in clone 5 and another in clone 6. All identified driver mutations were sequenced much deeper (with an average of 549 total read counts per variant in the bulk sample) in WES compared to ST data (average 0.02 read per variant per spot; Supplementary Fig. 5), motivating our probabilistic approach for clone mapping.

Next, given the selected 11,461 cancerous spots, their cell counts estimated from H&E images, total and alternated read counts at identified mutations, and the reconstructed clone genotypes, we used Tumoroscope to correct the initial, estimated number of cells in each spot and deconvolute the transcriptomics mutation profiles in spots to obtain the proportions of the underlying clones. We conducted a comparison between the corrected and the initial counts of cells (Supplementary Fig. 6). Our analysis reveals that Tumoroscope

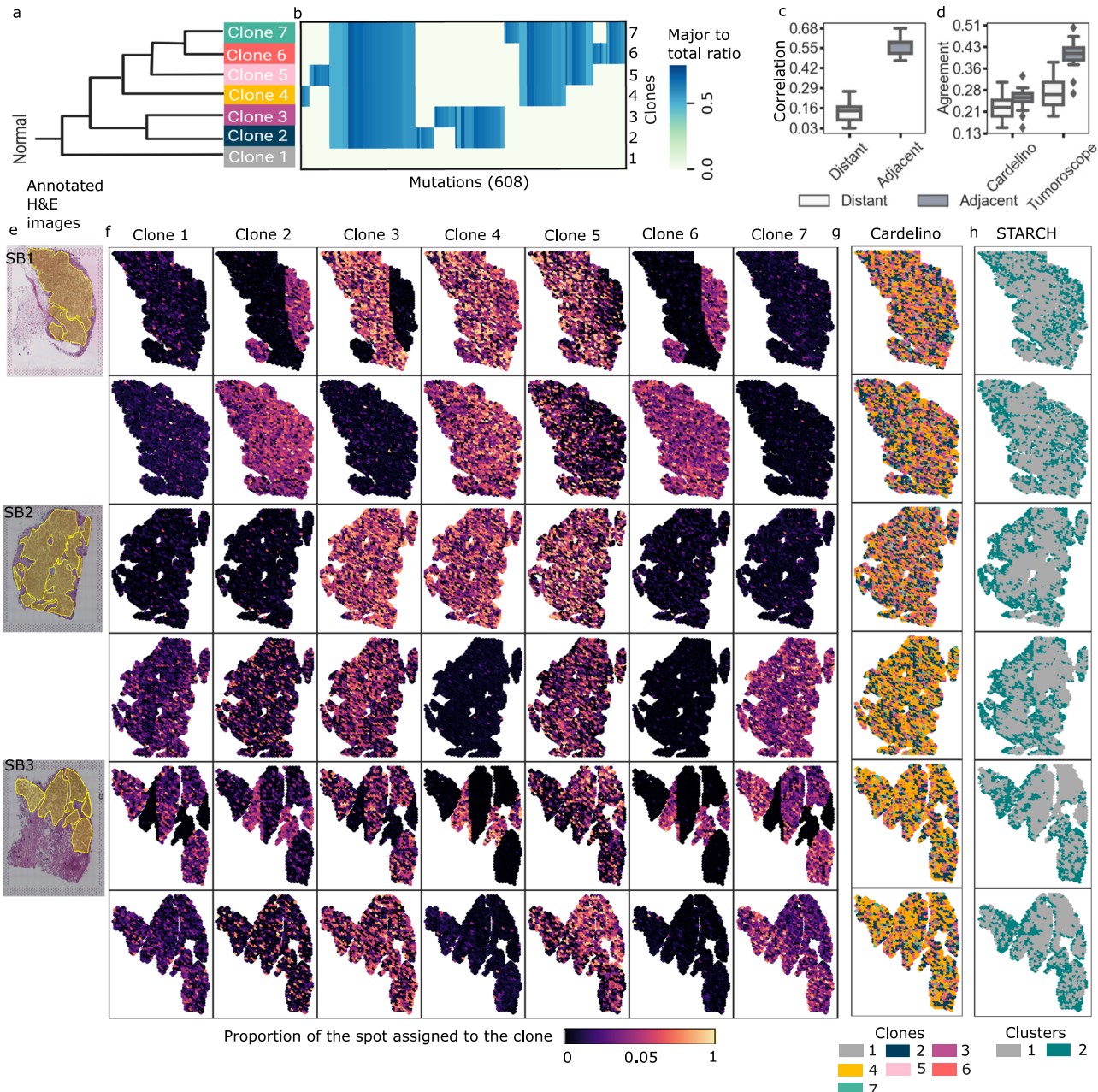

**Fig. 3 | Spatial arrangement of cancer clones inferred for the breast cancer dataset. a, b** Evolutionary tree and genotypes of the inferred clones. Figure 3a is created in BioRender[55]. Colors: major to total ratio, i.e., the fraction of the major copy number to the total copy number, with values that fall within the range of 0 to 1. **c** Distribution of the Pearson correlation (y-axis) of the clonal composition of the spots that are distant and adjacent, computed for 100 pairs of spots sampled at random 20 times each (x-axis). **d** Distribution of the agreement of the distant and adjacent spots in cardelino and Tumoroscope, computed for the same randomly sampled pairs as used in **c**. To compute the agreement, we use the single inferred clone by cardelino and the major inferred clone by Tumoroscope. In panel **c** and **d**, the lower and upper boundaries of the box represent the first (Q1) and third

quartiles (Q3), with the median indicated by a line inside the box. The whiskers typically extend to the most extreme data points within 1.5 times the interquartile range (IQR) from the quartiles. Data points outside this range are considered outliers and are plotted individually by diamonds. **e** Pathologist's annotation of the cancerous areas on the H&E images for sections SB1, SB2, and SB3. **f** For each section, two rows correspond to the two nearby samples and 7 columns correspond to the proportion of the spots assigned to each clone. **g** The clonal assignment of the spots inferred by cardelino for the same samples (see Supplementary Fig. 10 for expanded cardelino results). **h** Assignment of spots to copy number clones inferred by STARCH, with two clusters: gray corresponding to a normal clone, and dark blue corresponding to a single tumor clone.

exhibits a tendency to reduce the cell counts for spots with exceptionally high estimates, while simultaneously increasing counts for spots with notably low estimates. This behavior suggests that Tumoroscope is effectively addressing outliers, which could potentially represent errors in the initial estimates.

The composition of the seven clones in the investigated breast cancerous tissue identified by Tumoroscope revealed fascinating patterns of spatial arrangement (Fig. 3f). Generally, no single clone

fully dominated a specific contiguous sub-area of tissue. However, we did observe subsets of clones that coexisted within these sub-areas. For section SB1, clone 4 was present in medium proportions in all analyzed spots of both layers. Very interestingly, there was a clearly separated sub-area in the right-hand part of section SB1 first layer, where clones 2, 4, and 6 co-occurred. The rest of this layer was dominated by clones 3, 4, and 5. In the second layer of section SB1, clones 2, 4, 5, and 6 coexisted, although with larger proportions of

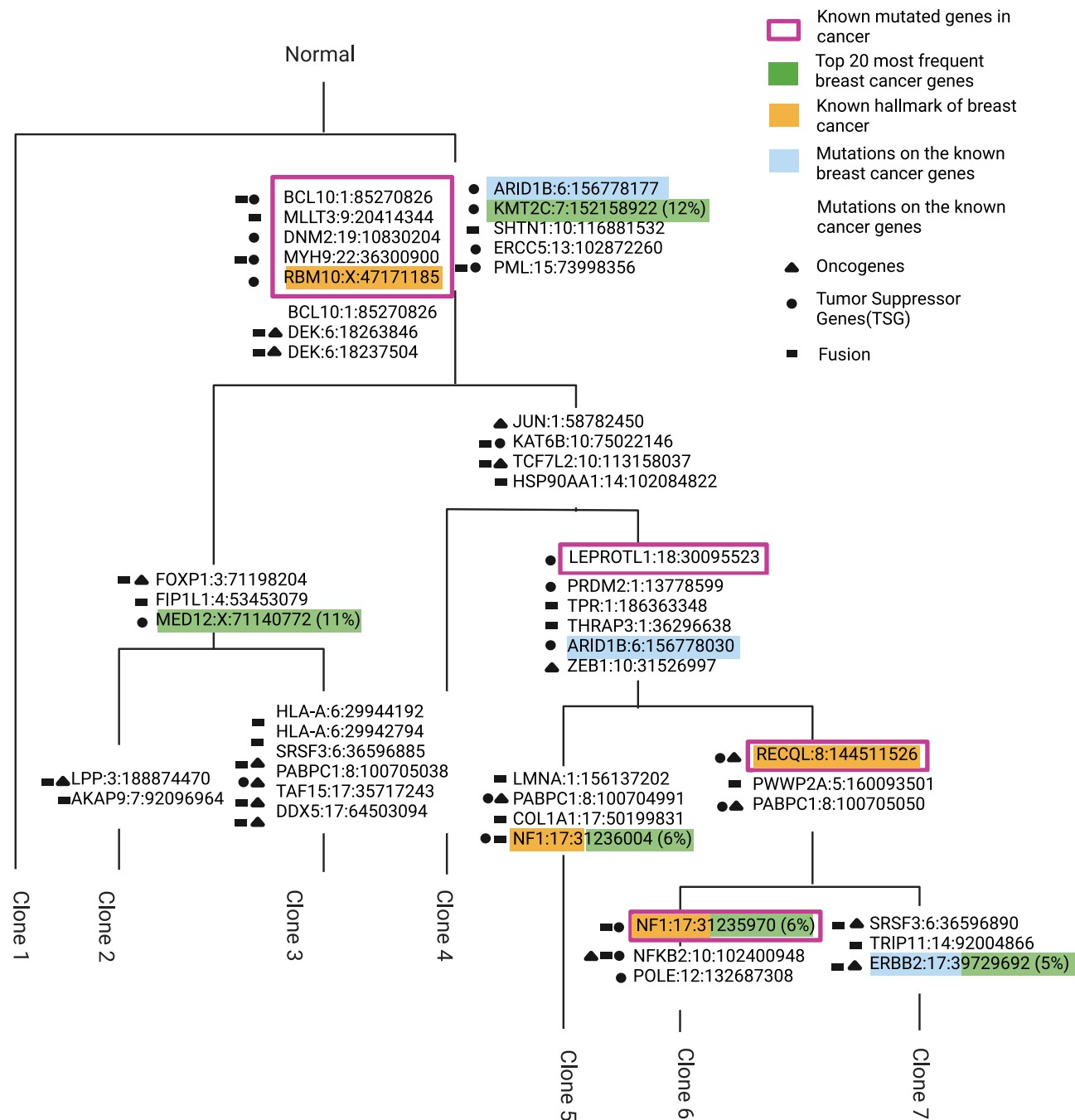

**Fig. 4 | Clonal evolution of breast cancer samples inferred by Canopy.** At each branch, known oncogenes, tumor suppressor or fusion genes with mutations that occurred along that branch are marked with black shapes. Colors: genes within 'Top 20 mutated genes in breast cancer' (green), 'Known hallmark of breast cancer' (yellow), and the 'Known breast cancer genes' (blue) categories. Purple framing: genes in the 'Known mutated genes in cancer' category. The branch lengths were adjusted for the visual presentation and are not inferred by the model. The percentage in the brackets for the 'Top 20 mutated genes in breast cancer' category is the mutation frequency (total mutated samples / total samples analysed, in percent) in the breast cancer samples in COSMIC. Figure is created in BioRender[56].

clone 4, low proportions of clones 2 and 6, and clone 5 being present in fewer spots than other clones. Clone 7 was not present in either layer of this section. Similarly, contiguous sub-areas that were predominantly occupied by small subsets of clones could be found in both layers of sections SB2 and SB3. As expected, clone 1, which lacked somatic mutations characteristic of the remaining cancerous clones, was found in only small proportions in the analyzed spots across all sections and layers.

Patterns of clonal co-occurrence and mutual exclusion could be observed across all sections and layers, indicating a systematic mechanism. For example, the pairs of clones 2 and 6, as well as 3 and 5,

although evolutionarily distant and with different genotypes (Fig. 3a, b) were always present together in the same sub-areas, while clones 4 and 7 excluded each other. To further investigate the interesting pattern of co-occurrence of evolutionary distant clones, we traced the alternate read counts corresponding to mutations found in clone 2 but not in clone 6, and conversely, mutations present in clone 6 but absent in clone 2 in ST slides (Supplementary Fig. 7a). This analysis provides confirmation of co-occurrence of the alternated reads for the mutations specific to clone 2 and mutations specific to clone 6. Interestingly, the colocalization of clones 2 and 6 also coincides with areas of higher coverage (Supplementary Fig. 8a,b). This, however, does not

affect the identification of the clone location in Tumoroscope. Instead, the higher coverage is accounted for in the model by inferring higher inferred variant expression values (Φ variables) for the variants in these clones, which we also confirmed (Supplementary Fig. 9). Similarly, we investigated the second co-occurring pair of clones 3 and 5, as well as the mutually exclusive pair 4 and 7, visualizing clone-specific mutations for those pairs (Supplementary Fig. 7b and c, respectively). While Tumoroscope leverages both clone-specific and common mutations simultaneously, the presence of clone-specific mutations confirms that the data provides the signal needed for inference to the model.

In contrast to Tumoroscope, in the assignments of single clones to spots inferred by cardelino, there was no detectable spatial pattern of domination of clones in sub-areas, as all clones were present in all sections uniformly (Fig. 3g; Supplementary Fig. 10). Also, the overall proportions of inferred clones differed between cardelino and Tumoroscope (Supplementary Fig. 11). Again, this underscored the importance of spot deconvolution.

To validate the decomposition results in the absence of ground truth, we relied on the natural expectation that adjacent spots would exhibit high similarity in clonal composition due to the spatial growth process of the tumor. We found that the median correlation of clone proportions inferred by Tumoroscope between adjacent spots was significantly higher than the median correlation between distant spots (computed between 100 pairs of spots each, sampled at random 20 times; Fig. 3c). Since Tumoroscope treated each spot as independent and did not enforce any spatial similarities by design, this result strongly supports the correctness of the deconvolution of ST spots using Tumoroscope.

Next, we compared Tumoroscope's and cardelino's performances. Since cardelino was originally designed to analyze scRNA-seq data, when applied to ST data, it assigned only one clone to each spot. Thus, to enable the comparison, similarly as we did for simulated data, for every spot of interest, we determined the major clone (characterized by the highest proportion) indicated by Tumoroscope. We then computed the agreement for each out of 20 randomly sampled sets of adjacent and distant pairs of spots considered previously (Fig. 3d). The median fraction of adjacent pairs of spots with clonal assignment in agreement was much higher for Tumoroscope (0.41) than for cardelino (0.25). Moreover, the difference between the agreement for the distant and adjacent pairs was larger for Tumoroscope (distance between medians 0.14; one-sided Wilcoxon $p$-value: 1.9e-06) than for cardelino (distance between medians 0.03; one-sided Wilcoxon $p$-value: 0.063).

To investigate the level of spatial and genomic heterogeneity contributed by CNVs, we additionally performed an analysis of the same tissue using STARCH[23], with default parameter settings. STARCH is a statistical method used for inferring CNV-based clones and determining copy numbers within spatial spots. Notably, it incorporates spot positions to account for the similarity of nearby spots and assumes that the spots are homogeneous, i.e., that each contains only one clone. As per the authors' recommendations, we ranged the number of CNV-based cancerous clones from 1 to 8 (excluding the normal clone) and relied on the Silhouette score to determine the optimal clone number (Supplementary Fig. 12). By design, STARCH includes the normal clone to the identified cancerous clones. Remarkably, this analysis revealed the presence of only a single CNV-based clone, in addition to the normal clone (Fig. 3f,h). This finding suggests that, in contrast to the significant heterogeneity identified by Tumoroscope in SNV-based clones within breast cancer samples, there is a lack of heterogeneity in terms of CNV-based clones for this tumor.

### Tumoroscope maps ST spots to clonal populations in a prostate tumor sample

Subsequently, we employed Tumoroscope to analyze three prostate tumor sections from a single patient. For these sections, we had access to deep WES and ST data (custom arrays) from adjacent tissue layers[26]. Following our established protocol, from the original 968-1001 spots per sample, we identified a total of 294 cancerous spots within tumor regions delineated by an expert pathologist. We then enumerated the cells in spots based on H&E images (Fig. 5c), observing between 1 and 188 cells per spot. These cell counts were subsequently refined by Tumoroscope during the model inference stage (Supplementary Fig. 6; Methods).

Our analysis revealed 282 high-confidence somatic SNVs found in WES data that were also detected in ST data. Using Canopy[9], we constructed an evolutionary tree for the tumor based on the WES data for these shared SNVs. This analysis uncovered four distinct clones, including a base clone devoid of somatic mutations (Fig. 6; Fig. 5a,b; Supplementary Fig. 13). Further investigation identified 18 mutations in cancer driver genes listed in the COSMIC database, distributed across the tree's branches. Notably, we detected a mutation in *KLK2*, a gene associated with 'Known prostate cancer genes', which was found in clone 3 (Fig. 6). Consistent with our observations in the breast cancer dataset, the sequencing coverage for mutated sites in the WES data significantly exceeded that of the ST data in this prostate dataset (Supplementary Fig. 5).

Lastly, we employed Tumoroscope to deconvolve the transcriptomic signals from 294 spots in the ST data, elucidating the proportions of underlying clones. As previously observed for breast cancer, we found a mosaic pattern of sub-regions with distinct clonal compositions (Fig. 5d). Notably, section SP1 exhibited a clear dichotomy: the left portion contained a mixture of cancer clones 2, 3, and 4, while the right portion was predominantly populated by clone 4, with a minor presence of normal cells (clone 1). Although smaller in size, sections SP2 and SP3 also displayed discrete sub-regions characterized by varying clonal compositions.

For comparison, we again applied cardelino, by considering each spot in the ST data as a single cell (Fig. 5e; Supplementary Fig. 14). Interestingly, similarly to Tumoroscope, for section SP1 cardelino also divided the tissue into two different subareas, confirming their distinct clonal composition. However, the clones assigned by cardelino did not agree with the clones identified as taking the most proportion of the same spots by Tumoroscope. For example, for the right-hand sub-area of section SP1, cardelino mostly assigned spots to clone 3, and not 4.

We further verified whether Tumoroscope inferred more similar clonal profiles for adjacent spots than for distant spots. As expected, the correlations of the inferred clone proportions between adjacent spots (median 0.65) were significantly higher than the correlations between distant spots (median 0.38; computed for 100 randomly selected pairs each and sampled 20 times; Fig. 5f).

Furthermore, we compared the percentage of the agreement of the major clone in each spot in the adjacent and distant pairs of spots found using Tumoroscope, with the agreement of the clones in the same pairs of spots assigned by cardelino (Fig. 5g). With a median of 0.41, the agreement for adjacent spots was significantly higher for Tumoroscope than for cardelino (median 0.31). Furthermore, the difference between the agreement of the adjacent and distant spots was significant for Tumoroscope (difference between medians 0.05; one-sided Wilcoxon $p$-value 0.004) and was notably greater than observed for cardelino (0.01; one-sided Wilcoxon $p$-value 0.556).

To compare SNV-based clonal heterogeneity inferred by Tumoroscope to CNV-based heterogeneity, we additionally utilized STARCH to deduce CNV-based clones within the cancerous prostate tissues. We conducted runs for 1 to 8 clones and, following the recommendations outlined in the paper, utilized the Silhouette score to ascertain the optimal clone number (Supplementary Fig. 12). Interestingly, STARCH failed to find tumor clones for nearly half of the spots and identified them as normal. For the same spots Tumoroscope successfully determined SNV-based clones. Here, Tumoroscope agreed with H&E data, which clearly indicated that the vast majority of

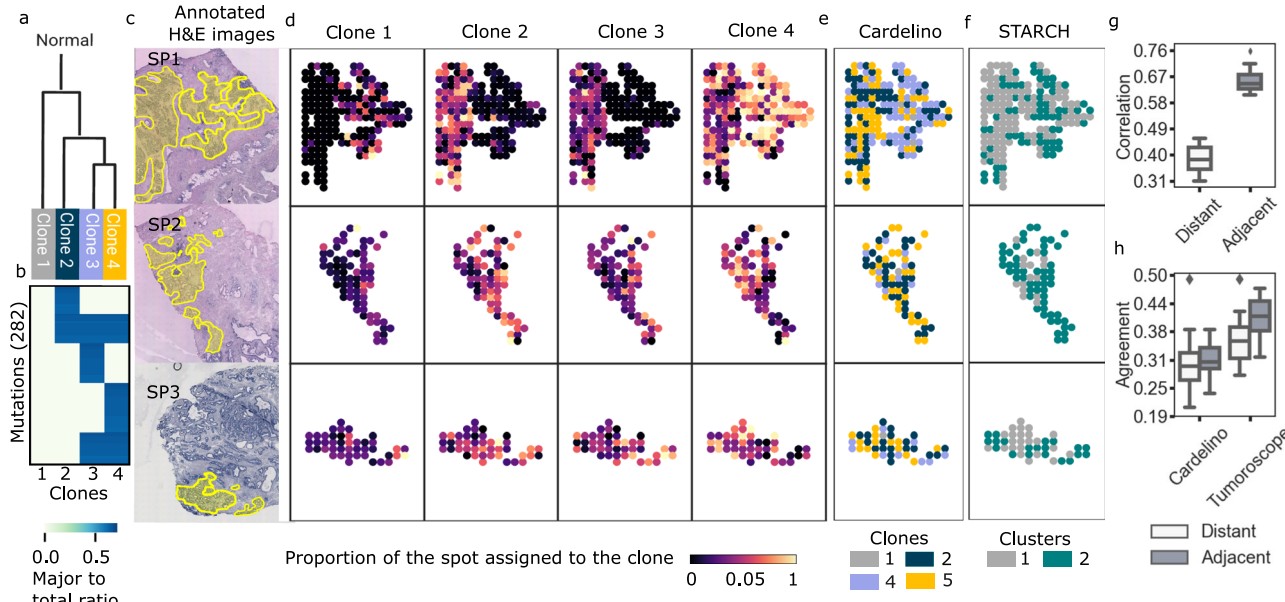

**Fig. 5 | Results obtained for the prostate cancer dataset. a-b** Evolutionary tree and genotype of the clones. Figure 5a is created in BioRender[57]. Colors: major to total ratio, i.e., the fraction of the major copy number to the total copy number, with values that fall within the range of 0 to 1. **c** Pathologist's annotation of the cancerous areas on the H&E images for sections SP1, SP2, and SP3. **d** For each section (rows), 4 columns correspond to the proportion of the spots assigned to each clone. **e** The clonal assignment by cardelino (see Supplementary Fig. 14 for expanded cardelino results). **f** Assignment of spots to copy number clones as inferred by STARCH, with two clusters: gray corresponding to a normal clone (with no copy number changes), and dark green corresponding to a single tumor clone. **g** Distribution of the Pearson correlation of the clonal composition of the spots that are distant and adjacent, computed for 100 pairs of spots sampled at random 20 times each. In panel **g** and **h**, the lower and upper boundaries of the box represent the first (Q1) and third quartiles (Q3), with the median indicated by a line inside the box. The whiskers typically extend to the most extreme data points within 1.5 times the interquartile range (IQR) from the quartiles. Data points outside this range are considered outliers and are plotted individually by diamonds. **h** Distribution of the agreement (y-axis) of the distant and adjacent spots for cardelino and Tumoroscope, computed for the same randomly sampled pairs as used in **g**. For the computation of the agreement, we use the single inferred clone by cardelino and the major inferred clone by Tumoroscope.

cells in this region are indeed cancerous (full-size H&E images are available for inspection; see Data Availability section). The misclassification of cancerous tissue as normal by STARCH may be due to the presence of true subclones that have evolved exclusively through SNV events, making them undetectable by STARCH. Identification of only a single tumor clone in other parts of cancerous tissue by STARCH could be due to copy number events occurring early in tumor evolution, affecting all cells in those subclones and thus lacking CNV heterogeneity. In contrast, SNVs could have occurred later in the evolutionary process, leading to the formation of distinct clones found by Tumoroscope. It's important to emphasize that within SP1, STARCH clearly distinguished between clones on the right-hand and left-hand sides, in agreement with Tumoroscope (Fig. 5d,f).

**Spatial proximity of clones reflects similarities in gene expression patterns**

We then utilized a regression model (Methods) to deconvolve clone-specific gene expression profiles from the measured expression data in spots. This model posits that each spot's gene expression is a composite of clone-specific expressions, weighted by their Tumoroscope-inferred proportions and adjusted for the estimated cell count per spot. For both cancer types, we prioritized genes based on their maximum inferred expression values across clones, selecting the top 30 genes for analysis.

In the prostate cancer dataset, 9 of the 30 selected genes (*KLK2, KLK3, MSMB, TAGLN, SPON2, KLK4, PMEPA1, MYH11, AZGP1*) are known to be enriched in prostate cancer tissues (i.e., have elevated expression in prostate cancer according to the Human Protein Atlas; HPA;[30]), while 19 are upregulated across various cancer types. For breast cancer, we identified *TPRG1*, known to be enriched in breast cancer tissues, along with 25 genes upregulated in multiple cancer types.

The deconvolved expression profiles varied among genes and clones (Fig. 7). Notably, we observed genes highly expressed in specific clone subsets. In breast cancer, for instance, *MT-CO1* and *MT-CO3*, associated with promoting cancer phenotype[31,32], and *RPL19*, linked to poor patient survival[33], were predominantly expressed in clones 2, 6, and 7. In prostate cancer, *KLK3*, a prostate-specific antigen and key prostate cancer clinical biomarker[34], was active in clones 2 and 3. These findings suggest that individual cancer cell clones may have distinct roles in tumor progression and development.

Lastly, we clustered the clones based on their gene expression profiles (Fig. 7). Fascinatingly, in both cancer types, clones with similar inferred phenotypes, as evidenced by their clustered expression profiles, also exhibited spatial co-localization across the tissue (cf. Figs. 3 and 5).

In the breast cancer analysis, we observed a notable correlation between the proportions of clones 2 and 6 across spots (Pearson correlation $r = 0.64$; Supplementary Fig. 15). Given this correlation, the regression model was expected to yield similar gene expression profiles for these clones. Interestingly, clones 3 and 5, despite showing small negative correlation in their spot fractions (Pearson correlation $r = -0.28$; Supplementary Fig. 15), exhibited spatial proximity, co-occurring in adjacent spots (Fig. 3; average correlation of fractions in adjacent spots $r = 0.16$; Supplementary Fig. 16). This spatial arrangement was not inherently accounted for in the regression model's construction, yet these clones were inferred to have the second most similar expression profiles.

For the prostate cancer sample, clones 2 and 3, which displayed spatial co-localization in the tissue (Fig. 5), also demonstrated the highest similarity in their inferred gene expression profiles.

Notably, in both cancer types, the pairs of clones showing correlated spatial distribution and gene expression were not closely related in terms of their mutational profiles (Fig. 3b). Consequently, these

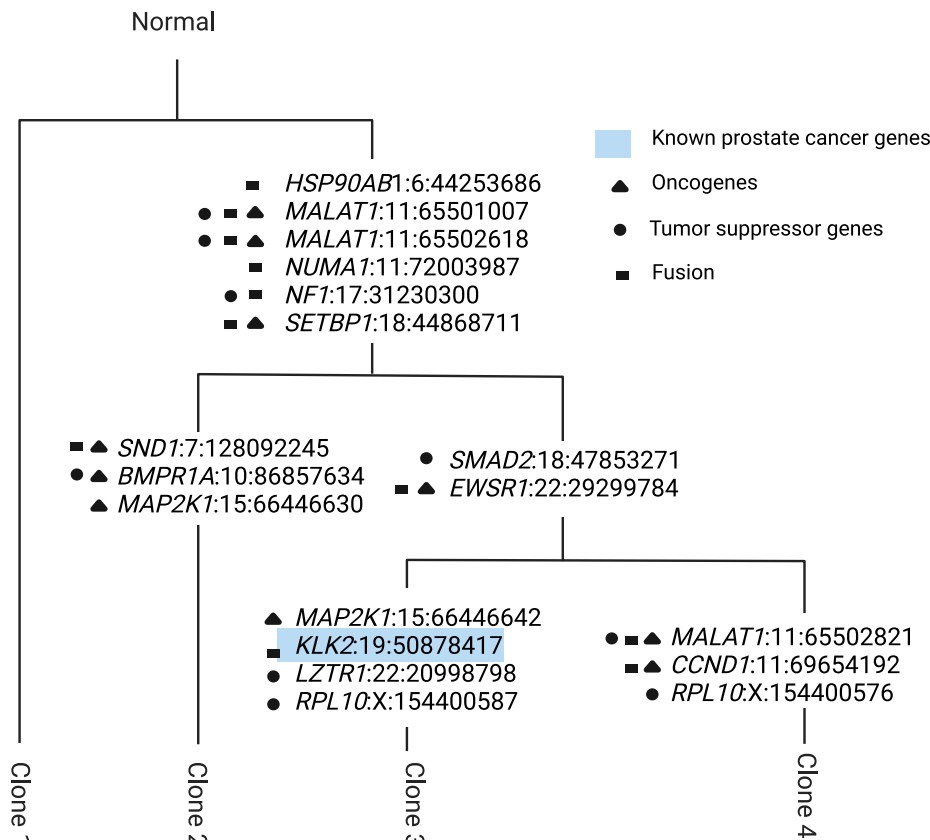

**Fig. 6 | Clonal evolution of prostate cancer samples inferred by Canopy.** At each branch, known oncogenes, tumor suppressor or fusion genes with non-synonymous mutations that occurred along that branch are marked with black shapes. Blue color marks the gene that belongs to the 'Known prostate cancer genes'. The branch lengths were adjusted for the visual presentation and are not inferred by the model. Figure is created in BioRender[58].

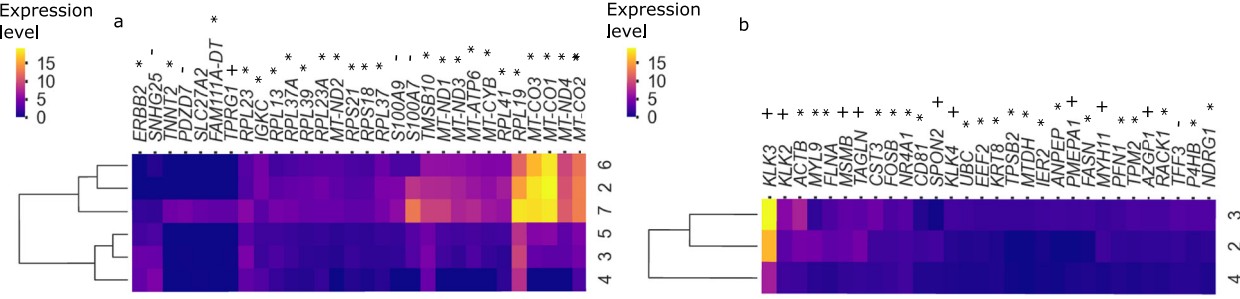

* cancer gene found in all cancer tissues [not type specific]
+ cancer gene with nTPM (Normalized gene expression values) in the desired cancer type at least four times higher than other cancer tissues.
++ cancer gene with nTPM in a group cancer tissues including the desired cacner type at least four times higher than any other cancer tissues.
- not detected in cancer tissues or nTPM is at least four times higher in another cancer tissue than the desired one

**Fig. 7 | Genes are expressed differently in various cancer clones.** The expression of the 30 genes that were inferred by the regression model as the most active in at least one clone, clustered in rows and columns, for breast (**a**) and prostate cancer (**b**) tissues. * cancer gene found in all cancer tissues (not cancer type specific) according to the HPA database[30]; + cancer gene with nTPM (normalized gene expression value) in the desired cancer type (either breast in **a** or prostate in **b**) at least four times higher than in other cancer tissues, according to[30]; ++ cancer gene with nTPM in a group of cancer tissues including the desired cancer type, at least four times higher than in other cancer tissues, according to[30]; - not detected in cancer tissues, or nTPM at least four times higher in another cancer tissue than the desired one, according to[30].

clones were positioned in distinct branches of the evolutionary tree (Fig. 3a). This observation suggests that even clones that are distant in their evolutionary history may converge on similar phenotypes and potentially fulfill comparable roles in tumor progression.

To investigate whether Tumoroscope is negatively affected by the copy number events, we investigated whether our regression model infers expression levels in accordance with copy number

(Supplementary Figs. 17, 18 and 19). Indeed, we observed that for genes with higher copy number inferred by STARCH, Tumoroscope also tended to infer higher expression values. This clear trend further confirms that our model is not adversely affected by the occurrence of CNVs and can correctly infer clone-specific expression profiles irrespective of their copy number. However, this observed concurrence might be attributed to the relatively low influence of CNV events in

shaping the evolutionary trajectory of the analyzed prostate and breast cancer data. We hypothesize that if multiple CNV tumor clones were present, potentially leading to the divergence of SNV-based clones during evolution, the agreement between these two factors would likely be less pronounced.

## Phenotypic similarity of phylogenetically distant clones is confirmed by independent scRNA-seq data analysis

To further validate gene expression profiles inferred by Tumoroscope for the breast cancer samples, we analysed scRNA-seq data from the same patient and tissue sections. To this end, we selected the top 100 highly variable genes in the profiles inferred by Tumoroscope and analyzed their expression in 120 cells in the scRNA-seq dataset. The dimension of scRNA-seq expression vectors was reduced using Principal Component Analysis (PCA) and next clustered using the K-means algorithm. The Dunn Index measure was used to identify the optimal number of 3 scRNA-seq clusters (see Supplementary Fig. 20). Finally, to render the seven expression profiles inferred from ST data comparable with the expression measurements of individual cells in the scRNA-seq data, the centers of expression vectors were harmonized (Methods). Next, we mapped each of the seven inferred clone gene expression profiles to the closest out of the 3 scRNA-seq clusters in terms of Euclidean distance to its centroid.

As a result, clones 1, 2, 3, 4, 5, 6, and 7 were assigned to two of the three scRNA-seq clusters: 1, 2, 1, 1, 1, 2, and 2, respectively. Inspection of the clones' assignment to these three expression clusters gives compelling evidence for their phenotypic similarities. In particular, clones 2 and 6, which were shown to be distant phylogenetically but found by Tumoroscope as colocalized in breast tissue (Fig. 3) and having similar inferred expression profiles (Fig. 7), were assigned to the same scRNA-seq cluster 2. Similarly, clones 3 and 5 found as co-localized and similar in expression by Tumoroscope, were assigned to the same scRNA-seq cluster 1 (Supplementary Fig. 21). Thus, the similarity of expression profiles for these pairs of clones was confirmed by independent clustering of scRNA-seq data from the same tissue.

## Discussion

Tumoroscope is the first approach for mapping cancer clones based on point mutations in tissue space and resolving their expression profiles in close to single-cell resolution. This resolution amounts to the diameter of the deconvoluted spots, ranging from 100 $\mu m$ (for the prostate cancer dataset[26]) to 55 $\mu m$ (for the breast cancer dataset), depending on the ST technology. Effectively, this means the model is able to assign clone proportions for spatially resolved mini bulks of a typical range of 1-100 cells. Tumoroscope achieves this result by innovative integration of data from technologies that were not originally developed for this task: H&E, WES, and ST. The primary signal leveraged by Tumoroscope to identify the clonal composition of ST spots is the alignment of mutations present in the genotypes of the clones to the mutations found in the RNA sequencing of the spots. Moreover, the method estimates additional variables, such as the number of cells in each spot and the average expression of each variant site per single cell. Finally, with the inferred proportions of the spots coming from specific clones, alongside the gene expression observed in spots in hand, we solve the problem of clone-specific gene expression deconvolution using a dedicated regression model.

Our comprehensive study on simulated data demonstrates Tumoroscope's robustness to noise in the estimation of the number of cells in ST spots. The results clearly indicate that the deconvolution task becomes easier with increasing coverage of mutations in ST spots and with a decreasing number of coexisting clones in each spot. In application to breast and prostate cancer data, Tumoroscope reveals spatial patterns of clonal arrangement, indicating a well-mixed coexistence of small subsets of all clones in subareas of the tumor tissue.

Applying our regression model to infer gene expression profiles in the different clones allows us to identify the distinct phenotypes of the clones, effectively assigning spatial resolution to the function of the different tumor subpopulations, and thus profiling the functional heterogeneity of tumors. Moreover, our findings in both analyzed cancer types indicate that it is the phenotypic, and not genotypic similarity, which could drive the spatial co-occurrence of clones. However, this result should be further validated in additional patient samples and using independent data.

To the best of our knowledge, no existing technology matches Tumoroscope's capacity to resolve spatial clonal heterogeneity in tumors at a comparable resolution. Spatial capturing of DNA sequences is still at the very early stage of development[35], with current methods yielding low-resolution data that necessitates merging adjacent beads, effectively producing spatial mini-bulk data similar to ST spots. Moreover, approaches that disregard the evolutionary origins of clones and cluster beads without considering variant allele frequencies, as performed in[35], oversimplify the complex task of spatial clonal deconvolution.

Given the inherent challenges of spatial DNA-seq data, high-resolution ST data emerges as a compelling alternative for spatially inferring clonal evolution. Recent methodologies like STARCH[23] combine ST RNA-sequencing with DNA-sequencing from adjacent tumor tissues to deduce clonal spatial arrangements based on copy number profiles. Similarly, Erickson et al.[24] developed a technique to infer copy number variations (CNVs) from spatially resolved in situ mRNA profiles, revealing distinct CNV-based clonal patterns within tumors. However, unlike Tumoroscope, these methods do not account for the impact of point mutations on tumor heterogeneity.

Our comparative analysis of STARCH and Tumoroscope results on breast and prostate cancer data underscores the critical importance of considering clones formed by point mutations. CNV-based models prove inadequate when clones arise solely from SNV events or when copy number alterations occur early in tumor evolution, resulting in their presence across all cells and minimal spatial CNV heterogeneity. Furthermore, current CNV-based models lack spot deconvolution capabilities, potentially compromising their performance when multiple clones coexist within individual ST spots.

Future technological advancements could further enhance the quality of results obtained through our approach. For instance, substituting WES with scDNA-seq data would enable more precise inference of cancer clones, their evolutionary relationships, and genotypes using dedicated computational methods[36,37]. Also, as ST technology progresses, we anticipate smaller spots, which would limit the number of clones per spot and simplify the deconvolution process.

Currently, Tumoroscope depends on external methods to provide clone information as input. This allows for the application of new state-of-the-art SNV clonal inference models as they become available. However, given the potential inaccuracies in such models, a promising avenue for improvement lies in directly incorporating the clonal inference within the model itself.

Additionally, while in the current Tumoroscope framework pathologists initially identify cancerous spots, the possibility of immune cell infiltration in some spots highlights the need for a more refined approach to preprocessing and cancerous spot selection. Theoretically, the model could address this by classifying such spots as belonging to the 'normal clone', as immune cells infiltrating solid tumors typically lack mutations.

At present, only the first 300 bp of gene sequences are sequenced in ST data generation. Ideally, whole gene bodies should be sequenced to allow all mutations detectable from WES to be observed in ST, enabling more accurate deconvolution of spots into clones. While such sequencing has recently been demonstrated[38], it was not available for the data we analyzed and is not part of standard ST protocols.

Despite these technological limitations, Tumoroscope represents a significant advancement in the integrated analysis of spatial, genomic, and phenotypic tumor heterogeneity through its novel fusion of data and modeling. It provides unprecedented insights into clonal composition, mutational genotypes of the clones, and their gene expression patterns in tissues, surpassing the capabilities of existing experimental or computational methods in terms of spatial, genetic, and phenotypic characterization.

The model's potential extends to future studies profiling adjacent tumor samples, enabling the creation of 3D clonal maps. By computing gene expression profiles of clones, we could predict the most proliferative areas and likely expansion sites within the 3D tumor structure. Furthermore, large-scale studies combining H&E, WES, and ST data across patient cohorts could explore relationships between clinical features and spatial clonal patterns identified by Tumoroscope.

When integrated with cell-type deconvolution approaches for ST data in the tumor microenvironment[39–41], our framework has the potential to provide unprecedented insights into the interactions between specific cancer clones, their phenotypes, and the surrounding microenvironment. In conclusion, Tumoroscope opens up new avenues in cancer research, with broad applications for both fundamental understanding of the disease and its clinical applications.

## Methods

### Breast cancer samples
The breast tumor samples originating from an untreated invasive ductal carcinoma patient, specifically identified as HER2 positive, were obtained from the Department of Clinical Pathology and Cancer Diagnostics at Karolinska University Hospital, Stockholm, Sweden. Experimental procedures and protocols were approved by the regional ethics review board (Etikprövningsmyndigheten) in Stockholm (2016/957-31, amendment 2017/742-32, 2021-00795, and 2022-05245-02). Before surgery, informed consent was given to the patient for signature. The extracted tissue was promptly embedded in OCT for preservation, facilitating subsequent gene expression analysis through spatial transcriptomics. The residual material from each tumor section was allocated for comprehensive whole exome sequencing and single-cell RNA sequencing analysis[42].

### ST experiments for breast cancer samples
For the three breast cancer sections, H&E images and ST datasets were generated as detailed by Engblom et al.[43]. Specifically, each section was extracted from a different region of the tumor, and for each section, two nearby samples were collected. Single samples from sections SB2 and SB3 were already analyzed by Engblom et al.[43], and the corresponding gene expression measurements were published there. Data for all six samples analyzed here are referenced in the Data and Code Availability section.

### Whole-exome sequencing for breast cancer samples
Concurrently, bulk DNA-seq procedures, as elucidated by Jun et al.[42], were conducted on closely adjacent samples for each section of the breast samples. Specifically, the collected tissues were manually homogenized and genomic DNA samples were isolated by using the QIAamp DNA mini kit (QIAGEN). The library was prepared by using Twist Bioscience Human Core Exome kit (Twist Bioscience) according to the manufacturer's protocol. The bulk DNA samples were then sequenced at 300x coverage depth in an S4 flow cell lane by the NovaSeq 6000 platform (Illumina) at the National Genomics Infrastructure, Science for Life Laboratory, Uppsala. Data for the WES samples is referenced in the Data and Code Availability section.

### ScRNA-seq experiments for breast cancer samples
The collected breast cancer samples were prepared for single-cell RNA sequencing as described by Jun et al.[42]. Next, for scRNA-seq libraries, the Smart-Seq3 method was used according to the published protocol (PMID: 32518404). The list of used oligonucleotides is provided as supplementary data of Jun et al.[42]. Data for the scRNA-seq is referenced in the Data and Code Availability section.

We employed a 384-well plate for single-cell RNA sequencing. Out of the total 384 cells, 224 were collected from the ST sections utilized in this study. We filtered the data based on the following criteria: mitochondrial expression fraction < 0.1, gene expression sum > 10,000, and the number of captured genes > 3,000, resulting in a final dataset of 120 high-quality cells for subsequent analysis.

### Prostate cancer samples
The prostate cancer dataset was generated and published by Berglund et al.[26]. This dataset consists of twelve sections, with H&E images, bulk DNA-seq, and spatial transcriptomics provided for each section. The data were generated and processed using protocols as described in[26].

### Identifying the spots that contain tumor cells
To select the spots that contain tumor cells, we took advantage of H&E staining images of the analyzed tissues. For both breast and prostate cancer, regions containing cancer cells were annotated by an expert pathologist, Dr. Łukasz Koperski, using QuPath[27]. We further selected spots whose area overlapped with the pathologist's annotated regions, using a custom script in QuPath[27].

### Counting cells in spots
We developed a custom script in QuPath[27] to count cells in each ST spot visible in the H&E images[27]. The script takes as input coordinates and diameters of spots to define target areas. Then, we employ QuPath's built-in cell counting algorithm to detect and count nuclei. In order to adjust parameters of the algorithm, we examined random spots by manually counting cells to verify the accuracy of the results.

### Spatial transcriptomics data preprocessing
For prostate cancer samples, the ST data bam files were provided by Berglund et al.[26]. For breast cancer samples, to create the genome index, we used the STAR program[44] with the GRCh38 reference genome as input. Next, we applied the ST Pipeline with default setting[45], providing the genome index, FASTQ files, barcodes and array coordinates as input. This pipeline ensures data integrity through quality checks, artifact removal, and genome alignment. We obtained the gene expression matrix as counts of reads for each gene, which the ST Pipeline produces by default. In addition, we modified the default settings to obtain bam files with the mapped reads.

### Bulk DNA-seq and somatic mutation calling
We identified somatic mutations that appeared in at least one of the bulk DNA-seq samples, by calling the mutations using Vardict[28] for each sample with a $p$-value threshold equal to 0.1. Then, we used their union over sections as the set of mutations called in bulk DNA-seq data. This procedure was performed in the same way for the prostate and the breast dataset.

### Selection of somatic mutations that are detected both in bulk DNA-seq and ST data
Next, we identified the bulk DNA-seq mutations that were also present in ST data with the same alternated nucleotide. To calculate the total and alternated reads over the mutations in ST data, we located the selected bulk DNA-seq mutations in the ST bam files and counted the corresponding mapped reads with our script. The reads with a different nucleotide as compared to the reference genome were called the alternated reads. For ST data, we use the terms spot coverage and mutation coverage to refer to the number of reads found in a spot or over a mutation, respectively.

Finally, we selected the mutations for which there existed at least one alternated read in at least one section. The alternated and total read counts in bulk DNA-seq data for the selected mutations were given as input for phylogenetic inference, while the alternated and total read counts in ST data for the same mutations were given as input to Tumoroscope. The median mutation coverage for the selected variant sites in bulk DNA-seq data for breast and prostate cancer were 214.5 and 134.75, respectively.

## Phylogenetic tree analysis

To identify the phylogenetic tree and infer the genotype and prevalence of each clone in the tree, we used a statistical method called Canopy[9]. The input to Canopy are variant allele frequencies of somatic single nucleotide alterations (SNAs), along with allele-specific mutation coverage ratios between the tumor and matched normal sample for somatic copy number alterations (CNAs). We used FalconX for producing the allele-specific mutation coverage ratio between tumor and normal sample[29]. We used the multi-sample feature of Canopy to infer the clonal evolution across the sections for both prostate and breast datasets.

## Mapping fractions of cells in ST spots to cancer clones using Tumoroscope

Tumoroscope is a probabilistic graphical model for estimating proportions of cancer clones in ST spots given alternated and total read counts over the analysed somatic mutations, genotypes and frequencies of the clones, and estimated cell counts per each spot (Fig. 1f). Let $i \in \{1, ..., M\}$ index the selected mutation positions, identified both in bulk DNA-seq and ST data. We are given a set of $K$ cancer clones, indexed by $k \in \{1, ..., K\}$ as input, which has been derived from bulk DNA-seq data. The genotypes of the input clones are represented as a matrix $\mathbf{C}$ with entries between 0 and 1 corresponding to the zygosity. $\mathbf{C}_{i,k}$ equals 0 if there is no mutation on position $i$ in clone $k$, equals 1 in case all alleles of that position carry the mutation, and equals 0.5 when the half of the alleles of that position carry the mutation. Note that there can be multiple alleles for position $i$. In general, the zygosity is defined as the ratio of the number of mutated alleles to the total number of alleles and we estimate it by the ratio of the major allele frequency to the total read count. The prevalence of the clones in the bulk DNA-seq is represented by the vector $\mathbf{F} = (\mathbf{F}_1, ..., \mathbf{F}_K)$, with values summing up to one. Let $s \in \{1, ..., S\}$ index the spots. We use a feature allocation model to account for the presence of clones in spots[46]. Specifically, we define $\mathbf{Z}_{s,k} \in \{0, 1\}$ as an indicator of the presence of clone $k$ in spot $s$. We assume a Bernoulli distribution over $\mathbf{Z}_{s,k}$ and a Beta prior over its parameter $\boldsymbol{\Pi}$ with hyper-parameter $\zeta_s$:

$$\mathbb{P}(\mathbf{Z}_{s,k}|\boldsymbol{\Pi}_{s,k}) \sim \text{Bern}(\boldsymbol{\Pi}_{s,k}), \quad (1)$$

$$\mathbb{P}(\boldsymbol{\Pi}_{s,k}|\zeta_s, K) \sim \text{Beta}\left(\frac{\zeta_s}{K}, 1\right). \quad (2)$$

Let $\mathbf{1} = \{1\}^K$ denotes a $K$-dimensional vector with all elements equal to 1. Bearing in mind the assumption about Beta prior over $\boldsymbol{\Pi}_{s,k}$, we calculate the expected number of nonzero entries in each spot $\mathbb{E}[\mathbf{Z}^T_{\cdot,k}\mathbf{1}]$ using the formula for the mean of the Beta distribution as[47,48]

$$\mathbb{E}[\mathbf{Z}^T_{\cdot,k}\mathbf{1}] = \sum_{k=1}^{K} \mathbb{E}[\mathbf{Z}_{s,k}] = K\,\mathbb{E}[\mathbf{Z}_{s,k}] = K\frac{\frac{\zeta_s}{K}}{\frac{\zeta_s}{K}+1} = \frac{K\zeta_s}{\zeta_s + K}. \quad (3)$$

Given this formula and the number of the clones $K$, we are able to control the expected number of clones in each spot by tuning shape parameter of the beta distribution, $\frac{\zeta_s}{K}$.

Our main goal is to estimate the proportions of clones in the spots, which are represented by the variable $\mathbf{H}$, a matrix with $S$ rows

and $K$ columns. The value of an element $\mathbf{H}_{s,k}$ is the fraction of spot $s$ coming from clone $k$. We consider a Dirichlet distribution over $\mathbf{H}_{s,\cdot} = (\mathbf{H}_{s,1}, ..., \mathbf{H}_{s,K})$,

$$\mathbb{P}(\mathbf{H}_{s,1}, ..., \mathbf{H}_{s,K}|\mathbf{F}', F_0, \mathbf{Z}_{s,\cdot}) \sim \text{Dirichlet}(\mathbf{F}_1'^{\mathbf{Z}_{s,1}}F_0^{1-\mathbf{Z}_{s,1}}, ..., \mathbf{F}_K'^{\mathbf{Z}_{s,k}}F_0^{1-\mathbf{Z}_{s,k}}). \quad (4)$$

Here, $F_0$ corresponds to a "pseudo-frequency", and results in non-zero proportions for all clones for each spot. We set $F_0$ to a small number, effectively assigning small proportions to clones that are not present in the spot. The $\mathbf{F}' = (\mathbf{F}_1', ..., \mathbf{F}_K')$ are obtained as discretized frequencies $\mathbf{F}$. Specifically, we discretize the values of $\mathbf{F}$ by dividing the range from 0 to 1 into 20 equal-sized bins and then round up the values to the upper-bounds of the bins and scale them by multiplicative factor $l$

$$\mathbf{F}_k' = l \times \frac{\lceil 20 \times \mathbf{F}_k \rceil}{20}, \quad (5)$$

where we used $l = 100$, but it can be specified by the user.

To sample $H$, we take advantage of the relation between Dirichlet and Gamma distribution[49] and draw $K$ independent random samples $(\mathbf{G}_{s,1}, ..., \mathbf{G}_{s,K})$ from $K$ Gamma distributions,

$$\mathbb{P}(\mathbf{G}_{s,k}|\mathbf{F}_k', F_0, \mathbf{Z}_{s,k}) \sim \text{Gamma}(\mathbf{F}_k'^{\mathbf{Z}_{s,k}}F_0^{1-\mathbf{Z}_{s,k}}, 1), \quad (6)$$

and then we calculate the proportions $\mathbf{H}$:

$$\mathbf{H}_{s,k} = \frac{\mathbf{G}_{s,k}}{\sum_{l=1}^{K} \mathbf{G}_{s,k}}. \quad (7)$$

The total read count at position $i$ in spot $s$ is represented by observed variable $\mathbf{D}_{i,s}$. We assume a Poisson distribution over $\mathbf{D}_{i,s}$,

$$\mathbb{P}(\mathbf{D}_{i,s}|\mathbf{H}_{s,\cdot}, \boldsymbol{\Phi}_{i,\cdot}, \mathbf{N}_s) \sim \text{Pois}\left(\mathbf{N}_s \sum_k \mathbf{H}_{s,k}\boldsymbol{\Phi}_{i,k}\right), \quad (8)$$

where $\boldsymbol{\Phi}_{i,k}$ is the average mutation coverage for the position $i$ across the cells from clone $k$, and $\mathbf{N}_s$ is the number of cells in spot $s$. The variables $\mathbf{N}_s$ can be fixed to a priori known values.

However, in most practical applications, the number of cells per spot is not known. This gives a compelling reason to estimate them as a part of model inference. We assume a Poisson distribution over $\mathbf{N}_s$,

$$\mathbb{P}(\mathbf{N}_s|\boldsymbol{\Lambda}_s) \sim \text{Pois}(\boldsymbol{\Lambda}_s), \quad (9)$$

where $\boldsymbol{\Lambda}_s$ is the expected number of cells in spot $s$. Also, we assume a Gamma distribution over $\boldsymbol{\Phi}_{i,k}$,

$$\mathbb{P}(\boldsymbol{\Phi}_{i,k}|r, p) \sim \text{Gamma}(r, p), \quad (10)$$

where $r$ and $p$ are the shape and rate hyperparameters, respectively.

$\mathbf{A}_{i,s}$ represents the number of alternated reads for position $i$ in spot $s$. We assume a Binomial distribution over $\mathbf{A}_{i,s}$,

$$\mathbb{P}(\mathbf{A}_{i,s}|\mathbf{D}_{i,s}, \mathbf{H}_{s,\cdot}, \boldsymbol{\Phi}_{i,\cdot}, \mathbf{C}_{i,\cdot}) \sim \text{Binom}\left(\mathbf{D}_{i,s}, \frac{\sum_{k=1}^{K}\mathbf{H}_{s,k}\boldsymbol{\Phi}_{i,k}\mathbf{C}_{i,k}}{\sum_{k=1}^{K}\mathbf{H}_{s,k}\boldsymbol{\Phi}_{i,k}}\right). \quad (11)$$

Where the success probability of Binomial distribution is the probability of observing $\mathbf{A}_{i,s}$ alternated reads out of $\mathbf{D}_{i,s}$ reads in total. Given the variables $\mathbf{N}_s$, $\mathbf{H}_{s,\cdot}$ and $\boldsymbol{\Phi}_{i,\cdot}$, we calculate the expected number of alternated reads and the total reads in spot $s$ using $\mathbf{N}_s \sum_{k=1}^{K}\mathbf{H}_{s,k}\boldsymbol{\Phi}_{i,k}\mathbf{C}_{i,k}$ and $\mathbf{N}_s \sum_{k=1}^{K}\mathbf{H}_{s,k}\boldsymbol{\Phi}_{i,k}$, respectively. Therefore, to calculate the success probability, we determine the fraction of expected alternative reads to the total reads, which remains constant despite fluctuations in gene expression levels.

## Metropolis-Hasting inside Gibbs Sampling

In the Gibbs sampling, we iteratively generate samples from each hidden variable's conditional distribution, given the remaining variables, in order to estimate the posterior distribution of the hidden variables. Each hidden variable given the variables in its Markov Blanket is conditionally independent of all variables outside its Markov Blanket in the graphical model[50]. A variable's Markov Blanket includes its parents, children, and children's parents. If the conditional distribution does not have a closed analytical form, we use a Metropolis-Hasting step inside the Gibbs sampler. In the following, we describe the sampling steps for each hidden variable.

## The variables with the closed-form sampling distribution

$\Pi_{s,k}$ and $Z_{s,k}$ are the only variables with analytical sampling distributions.

### Sampling $\Pi_{s,k}$

For sampling $\Pi_{s,k}$, we take advantage of the conjugacy of Beta and Bernoulli distributions:

$$\mathbb{P}(\Pi_{s,k}|\zeta_s, K, Z_{s,k}) \propto \mathbb{P}(\Pi_{s,k}|\zeta_s, K)\mathbb{P}(Z_{s,k}|\Pi_{s,k})$$
$$= \text{Beta}\left(\Pi_{s,k}|\frac{\zeta_s}{K}, 1\right)\text{Bern}(Z_{s,k}|\Pi_{s,k}) = \text{Beta}(\Pi_{s,k}|\frac{\zeta_s}{K} + Z_{s,k}, 2 - Z_{s,k}). \quad (12)$$

### Sampling $Z_{s,k}$

For sampling $Z_{s,k}$, we utilize the fact that this variable only accepts binary values. Therefore, we sample 0 or 1, proportional to their corresponding calculated probabilities.

$$\mathbb{P}(Z_{s,k}|\Pi_k, G_{s,k}, F_k, F_0, l) \propto \mathbb{P}(Z_{s,k}|\Pi_k)\mathbb{P}(G_{s,k}|F_k, F_0, l)$$
$$= \text{Bern}(Z_{s,k}|\Pi_k)\text{Gamma}(G_{s,k}|F_k^{Z_{s,k}}F_0^{1-Z_{s,k}}, 1). \quad (13)$$

## Metropolis-Hasting adaptive steps inside Gibbs sampler

In our model, there is no closed analytical form of conditional distribution for variables $\Phi_{i,k}$, $G_{s,k}$ and $N_s$. Therefore, we take advantage of Metropolis-Hasting inside Gibbs sampler. We compute the acceptance ratio $A$ as the following

$$A = \frac{f(x_c)Q(x_n|x_c)}{f(x_n)Q(x_c|x_n)}. \quad (14)$$

Where $f(x)$ is a function that is proportional to the desired density function $P(x)$ and $Q$ is the proposal distribution. Bearing in mind the non-negativity of the variables of our interest, we choose a Truncated Normal distribution for $Q$ with the mean value of the current sample $x_c$ and variances $\sigma_\Phi$, $\sigma_G$ and $\sigma_N$ corresponding to each variable. The variance of the Truncated Normal distribution determines the proximity of the new sample from the current one, which is interpreted as the step size. The choice of the step size has a major impact on the acceptance rate of the Metropolis Hasting. We tune the $\sigma_\Phi$, $\sigma_G$ and $\sigma_N$ every $b$ steps starting with an arbitrary value based on the feedback from the acceptance rate. Firstly, we choose an optimal acceptance rate $R_o$ for each variable. Secondly, we modify the variance by $\delta$ percent of the current variance, and $\delta$ is calculated by the difference between the optimal and current acceptance rate $R_c$. Ultimately, during the sampling steps, we learn the optimal variance value for each variable.

$$\delta_t = R_o - R_c \quad (15)$$

$$\sigma_{t+1} = \sigma_t(1 + \delta + t) \quad (16)$$

In the following, we describe the conditional distribution for each variable.

## Conditional distribution for $\Phi_{i,k}$

$$\mathbb{P}(\Phi_{ik}|r, p, A_{i,\cdot}, H_{\cdot,k}, C_{i,\cdot}, D_{i,\cdot}, N_\cdot)$$
$$\propto \mathbb{P}(\Phi_{ik}|r, p) \prod_s \mathbb{P}(A_{i,s}|H_{s,k}, \Phi_{i,k}, C_{i,k}) \prod_s \mathbb{P}(D_{i,s}|\Phi_{i,k}, H_{s,k}, N_s)$$
$$= \text{Gamma}(r, p)\prod_s \text{Binom}\left(A_{i,s}|D_{i,s}, \frac{\sum_{k=1}^K H_{s,k}\Phi_{i,k}C_{i,k}}{\sum_{k=0}^K H_{s,k}\Phi_{i,k}}\right) \prod_s \text{Pois}\left(D_{i,s}|N_s \sum_k H_{s,k}\Phi_{i,k}\right). \quad (17)$$

## Conditional distribution for $G_{s,k}$

$$\mathbb{P}(G_{s,k}|F_k', F_0, Z_{\cdot,k}, A_{\cdot,s}, D_{\cdot,s}, \Phi_{\cdot,k}, C_{\cdot,k}, N_s)$$
$$\propto \mathbb{P}(G_{s,k}|F_k, F_0, Z_{s,k}) \prod_i \mathbb{P}(A_{i,s}|H_{s,k}, \Phi_{i,k}, C_{i,k}) \prod_i \mathbb{P}(D_{i,s}|\Phi_{i,k}, H_{s,k}, N_s)$$
$$= \text{Gamma}(F_k'^{Z_{s,k}}F_0^{1-Z_{s,k}}, 1)\prod_i \text{Binom}\left(A_{i,s}|D_{i,s}, \frac{\sum_{k=1}^K H_{s,k}\Phi_{i,k}C_{i,k}}{\sum_{k=0}^K H_{s,k}\Phi_{i,k}}\right) \prod_i \text{Pois}\left(D_{i,s}|N_s \sum_k H_{s,k}\Phi_{i,k}\right). \quad (18)$$

## Sampling $N_s$

$$\mathbb{P}(N_s|\Lambda_s, D_{\cdot,s}, \Phi, H_{s,\cdot}) \propto \mathbb{P}(N_s|\Lambda_s) \prod_i \mathbb{P}(D_{i,s}|\Phi_{i,\cdot}, H_{s,\cdot}, N_s)$$
$$= \text{Pois}(N_s|\Lambda_s) \prod_i \text{Pois}\left(D_{i,s}|N_s \sum_k H_{s,k}\Phi_{i,k}\right). \quad (19)$$

## Parameter setting for different simulation setups

First, we calculate the parameter of the Beta distribution over variable $\Pi_{s,k}$ based on the assumed expected value of the number of clones:

$$\frac{\zeta_s}{k} = \frac{\mathbb{E}\left[Z_{\cdot,k}^T \mathbf{1}\right]}{K - \mathbb{E}\left[Z_{\cdot,k}^T \mathbf{1}\right]}. \quad (20)$$

Considering expected values of 1, 2.5, and 4.5 for the number of clones found in each spot, we obtain 0.25, 1, and 9 and use these values for the Beta distribution parameter.

Second, we exploit $\Phi_{i,k}$ that represents the expected number of reads for mutation $i$ in each cell for generating different read coverage for total and alternated read counts. We set $p = 1$. With this, we control the expected value of $\Phi_{i,k}$ using parameter $r$.

$$\mathbb{P}(\Phi_{i,k}|r, p) \sim \text{Gamma}(r, p), \quad (21)$$

$$\mathbb{E}[\Phi_{i,k}] = \frac{r}{p^2}. \quad (22)$$

For the very low, low, medium and high number of reads, we consider $r = 0.02$, $r = 0.07$, $r = 0.09$ and $r = 0.19$, respectively, leading to the 18, 50, 80, and 110 average total reads for each spot.

Last, we generate three datasets for the number of cells with different levels of noise to compare our two models, which have the number of cells as observed and hidden variables. We add the noise value $\epsilon$ to the true values.

$$N_s = N_s + \epsilon. \quad (23)$$

We consider $\epsilon = 0$, $\epsilon \sim \text{Pois}(1)$ and $\epsilon \sim \text{Pois}(10)$ for generating without noise, noisy and highly noisy number of cells.

## Parameter estimation obtained for the real data

For the higher accuracy of the graphical model reflecting the real data, we estimate the input parameters of the model based on the characteristics of the data. The first parameter is $\lambda_s$, the expected number of the cells in spot $s$, which affects the estimation of the number of cells and, ultimately, the number of reads we are expecting, which is a

crucial element for estimating the fraction of the clones. Therefore, we estimate the number of cells using the H&E images and a customized script in QuPath and use them as the mean parameter for the Poisson distribution over $N$(described above)[27]. Next parameters are $r$ and $p$, the shape and rate in the Gamma distribution over variable $\Phi$. We use mixed type log-moment estimators for calculating $r$ and $p$[51].

$$\hat{r} = \frac{I\sum_{i=1}^{I}\mathbf{x}_i}{I\sum_{i=1}^{I}\mathbf{x}_i\ln(\mathbf{x}_i) - \sum_{i=1}^{I}\ln(\mathbf{x}_i)\sum_{i=1}^{I}\mathbf{x}_i}. \quad (24)$$

$$\hat{p} = \frac{I^2}{I\sum_{i=1}^{I}\mathbf{x}_i\ln(\mathbf{x}_i) - \sum_{i=1}^{I}\ln(\mathbf{x}_i)\sum_{i=1}^{I}\mathbf{x}_i}. \quad (25)$$

Where $\mathbf{x}_i$ with $i \in \{1, ..., I\}$ are the sample from Gamma distribution. We generate these samples using the total number of reads $\mathbf{D}$. We calculate the average number of reads from every cell, dividing the reads from the spots by the number of estimated cells as input, which gives us $I$ samples, equal to the number of mutations.

$$\mathbf{x}_i = \frac{1}{S}\sum_s \frac{\mathbf{D}_{i,s}}{\mathbf{n}_s}. \quad (26)$$

### Clonal composition resemblance in adjacent spots

The evolutionary process imposes the similarity of the clonal composition in the adjacent spots. Therefore, we expect to have a higher correlation between the clonal composition of the adjacent spots as compared to distant spots. To make this comparison, we randomly generate $N$ pair of adjacent spots $([(\mathbf{X}_1, \mathbf{Y}_1), (\mathbf{X}_1', \mathbf{Y}_1')] \ldots [(\mathbf{X}_N, \mathbf{Y}_N), (\mathbf{X}_N', \mathbf{Y}_N')])$ with $\mathbf{X}$ and $\mathbf{Y}$ corresponding to their coordinates. These adjacent pairs satisfy two constraints of $\mathbf{X}_j - \mathbf{X}_j' \leq 1$ and $\mathbf{Y}_j - \mathbf{Y}_j' \leq 1$ indexed by $j \in \{1, ..., N\}$. We also generate $N$ pair of distant spots with the two constraints of $\mathbf{X}_j - \mathbf{X}_j' > 1$ and $\mathbf{Y}_j - \mathbf{Y}_j' > 1$. We define $[\mathbf{V}_{k,j}, \mathbf{V}_{k,j}']$ as the fraction of clone $k$ in spots corresponding to the $j^{th}$ pair in the adjacent spots. Then we calculate the Pearson correlation for the vector $[(\mathbf{V}_{k,1}, \mathbf{V}_{k,1}'), \ldots, (\mathbf{V}_{k,N}, \mathbf{V}_{k,N}')]$. The procedure is repeated for all the clones and distant spots for the sake of comparison.

### Clonal assignment of the spots using cardelino

Cardelino[14] is a statistical method originally developed for inferring the clone of origin of individual cells using scRNA-seq. It integrates information from imperfect clonal trees inferred from WES data and sparse variant alleles expressed in scRNA-seq data. However, here, we applied it to ST instead of scRNA-seq to validate the assumption of a mixture of clones in each ST spot instead of assuming homogenous spots containing only one clone. We used *clone_id* function with "sampling" inference mode, minimum iteration of 100000 and maximum iteration of 250000. We used 3 parallel chains for prostate cancer data and 1 chain for breast cancer data due to the high RAM demand of cardelino. The RAM demand of cardelino grows with the number of spots. With 294 spots in the prostate datasets and 11,461 spots in the breast cancer dataset, it is nearly 39 times larger and running 3 chains becomes computationally infeasible.

### Assigning spots to copy number clones using STARCH

To perform the ST data analysis using STARCH, we have used the default code configuration, as recommended by the authors. Specifically, we used the run command

```
python  run_STARCH.py  -i  gene_expression_matrix.csv
(or 10X_directory/)

--output name --n_clusters K --outdir output/directory/
```

where $K$ is the number of clusters (corresponding to the number of identified copy number clones). As suggested by the Authors, we selected the number K of clones by computing the average silhouette score for a range of K and selecting the value of K with the highest average silhouette score.

### Estimating gene expression of the clones

Having the proportions of the clones in each spot inferred using Tumoroscope and gene expression data from ST, we estimate average clonal gene expression using a regression model. Let $g \in \{1, ..., G\}$ index genes and $\mathbf{Y}$ be a matrix with $S$ rows and $G$ columns, where $\mathbf{Y}_{s,g}$ is the measured gene expression of gene $g$ in spot $s$. We are interested in estimating $\mathbf{B}_{k,g}$ - average gene expression of gene $g$ in one cell of clone $k$. We use $\mathbf{H}$ and $\mathbf{N}$ variables inferred by Tumoroscope, and we rewrite $\mathbf{N}$ as an $S \times S$ diagonal matrix $\mathbf{N}'$, where $\mathbf{N}_{s,s}'$ is the number of cells in spot $s$ and other elements of the matrix are equal to zero. We describe the relationship between the variables with an overdetermined system of equations $\mathbf{N}'\mathbf{H}\mathbf{B} = \mathbf{Y}$. Then we try to find the optimal solution of this equation using linear regression with a lower bound of $\mathbf{B}_{k,g} \geq 0$ and no intercept. For this purpose, we apply a Python function *scipy. optimize. lsq_linear* to the data.

### Validation of gene expression profiles via independent single-cell RNA-seq data

The scRNA-seq dataset comprised 302 cells and captured the expression of 24,300 genes. We selected the top 100 genes with the highest variance across the clone-specific gene expression profiles inferred by Tumoroscope.

To mitigate potential biases arising from the use of two distinct technologies for gene expression measurement, we harmonized the data by aligning the centers of both the scRNA-seq data and our inferred profiles. This alignment involved adding the distance between these two centers to the Tumoroscope-inferred profiles, followed by Z-score normalization on both cells and genes.

Subsequently, we employed PCA to project the data into a space of reduced dimension. Next, we performed K-means clustering on the reduced gene expression profiles of individual cells. To determine the optimal number of clusters for K-means, we tested a range from 5 to 25 clusters and evaluated each clustering solution using the Dunn Index measure.

### Reporting summary

Further information on research design is available in the Nature Portfolio Reporting Summary linked to this article.

## Data availability

The sequencing data for the prostate cancer samples are published by Berglund et al.[26] and deposited at the European Genome-Phenome Archive (EGA), hosted by the European Bioinformatics Institute (EBI), under the accession number EGAS00001003001. Berglund et al. referred to sections SP1, SP2, and SP3 as p3.3, p2.4, and p1.2. Sequencing data for all breast cancer samples analyzed here (SB1, SB2, and SB3, two samples each) are deposited at EGA by accession number EGAD00001011061. In this repository, the ST samples from SB1 section are referred to as '112_C1' and '112_D1', samples from SB2 section are referred to as '114_D1' and '114_C1', and samples from SB3 section are referred to as '113_A1' and '113_B1'. Single breast cancer samples from ST sections SB2 and SB3 are analysed and published by Engblom et al. in Zenodo[43,52]. The bulk DNA-seq data and single-cell RNA-seq data for the breast cancer data are published in Jun et al.[42] and deposited at the EGA with accession number EGAS00001006851. The processed datasets supporting the conclusions of this article, including the evolutionary tree, frequency of the clones, alternated and total read count for the subset of somatic mutations found both in bulk DNA-seq and ST data for both breast and prostate cancer samples, as well as the scRNA-seq

count data for breast cancer is available at https://github.com/szczurek-lab/Tumoroscope. Additionally, the simulated data presented in this paper, along with the high-resolution H&E images for breast cancer data are available in Zenodo[53]. Source data are provided with this paper.

## Code availability

Tumoroscope can be obtained as an installable Python package, via 'pip install tumoroscope', and is available under the GNU General Public License v3.0. Tumoroscope implementation and package updates will be maintained at https://github.com/szczurek-lab/Tumoroscope. The scripts for ST read counting are available at https://github.com/szczurek-lab/st_read_counter and the script for the cell counting using QuPath from annotated H&E images is available at https://github.com/szczurek-lab/qupath-spot-utils.

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

## Acknowledgements

We acknowledge Andrew Roth for suggesting the feature allocation model as part of Tumoroscope. We also acknowledge the support from the National Genomics Infrastructure in Stockholm and Uppsala, funded by Science for Life Laboratory, the Knut and Alice Wallenberg Foundation, the Swedish Research Council, and SNIC/Uppsala Multidisciplinary Center for Advanced Computational Science for assistance with massively parallel sequencing and access to the UPPMAX computational infrastructure. This project has received funding from the European Union's Horizon 2020 research and innovation programme under the Marie Skłodowska-Curie grant agreement No 766030 and 844712, the Polish National Science Centre PRELUDIUM grant no. 2021/41/N/ST6/03619, the Polish National Science Centre OPUS grant no. 2019/33/B/NZ2/00956, the Polish National Science Centre SONATA BIS grant no. 2020/38/E/NZ2/00305, The Swedish Cancer Society, The Institut Universitaire de France (AC) and the Swedish Research Council Projekt 2018-06217VR.

## Author contributions

S.S., E.S., J.L. and A.G. developed the probabilistic model. S.S. implemented the model and carried out the application of the model, supervised by E.S. B.J. applied the regression on gene expression data and performed GSEA analysis. J.H. extracted the breast cancer tumor. X.C. prepared the sample and performed the DNA extraction for whole-exome sequencing of breast cancer dataset. K.T. and C.E. performed spatial transcriptomics. J.E.M. and C.E. performed the cell sorting. A.S.N. performed st pipeline and bulk DNA-seq mutation calling. H.T. and S.S. conducted the pre-processing of scRNA-seq data for mutation calling and gene expression analysis. S.S. performed the post analysis of the scRNA-seq data. S.S., J.L. and E.S. conceived the study. S.S., E.S., A.C., A.G. and J.L. wrote the paper. S.S. carried out the benchmarking of alternative methods. Ł.K., A.R. and I.F. analysed the H&E images. S.S., A.C., D.N., Ł.K. analysed and interpreted the model results. All authors provided critical feedback; helped shape the research and analysis; edited, reviewed and approved the manuscript.

## Funding

## Competing interests

Projects in Szczurek lab are co-funded by Merck Healthcare. C.E., K.T., and J.E.M. are scientific consultants for 10x Genomics Inc. Other authors declare no competing interests.

## Additional information

**Peer review information** : *Nature Communications* thanks the anonymous reviewers for their contribution to the peer review of this work. A peer review file is available.

