## [Transparent Peer Review file · Nature Communications]

Integrative Spatial and Genomic Analysis of Tumor Heterogeneity with Tumoroscope

Corresponding Author: Professor Ewa Szczurek

Version 0:

Reviewer comments:

Reviewer #1

(Remarks to the Author)

The authors developed a new method, Tumoroscope, to map inferred clones in the tumor tissue for spatial transcriptomic (ST) data. Using bulk DNA sequencing data, it first infers clones by using an available clone prediction method, Canopy. Then it uses read counts from different spots (mini-bulk sequences) in the tissue to estimate the frequency of each clone at each spot. The approaches used are standard, but the application domain is important. There is relatively little by the way of innovation, even though I see room for much innovation here. The comparison with Cardelino using simulated datasets, which is not really designed for spatial transcriptomic data, is presented. This is not a very informative comparison because it is designed to map inferred clones to single cells using SNVs from single-cell RNA-seq data. In addition, the Tumoroscope estimates gene expression for each clone within a spot, which is used to analyze new and useful ST datasets from breast and prostate cancers.

1. Clone decomposition from a single bulk sample is extremely difficult empirically and theoretically. One cannot believe that the Canopy (the method used) can accurately infer clones, and the impact of errors in the clone prediction is expected to be large, which is a major weakness of this work.
2. It is unclear how Tumoroscope uses the information on CNAs? In Tumoroscope, Canopy is performed by giving the information of CNAs. I guess inferred clones from Canopy include the status of CNAs?
3. Clone frequency at each spot is estimated using observed read count within a spot. However, the read counts are obtained from RNA-seq. So, over-expression and under-expression of some genes at a spot will affect the observed read count. Also, some mutant alleles are not expressed (Househam et al., Nature. 2022. doi: 10.1038/s41586-022-05311-x). How will all this affect the performance of clone fraction (frequency) estimation in the Tumoroscope?
4. Other methods for spatial transcriptomic data use CNAs to identify clones, e.g., STARCH (lines 333-337). It seems CNAs are common in cancer. When data is affected by CNAs, will Tumoroscope perform better than the existing methods?
5. Also, was the empirical data affected by CNAs? If so, it needs to be discussed to interpret the results of the Tumoroscope, e.g., gene expression pattern for those affected by CNAs.
6. Scripts and settings to process ST and bulk data for Tumoroscope analysis should be available for others. Also, demonstration of some example data analysis starting from processing raw ST data will be useful for others to use Tumoroscope for their data.
7. An extensive simulation study was performed, so simulated data should be available. Importantly, biological justification for the selection of parameters used in the computer simulation is needed, particularly because biological datasets available are rather limited at present.
8. How was a mean average error (MAE) computed? MAE was defined as "the average of the difference between the inferred proportions of the clone and the true values in all the spots and clones." Specifically, how was inferred clone annotated to the true clone?

9. Fig. 1h-1l and lines 151-153: How was the accuracy computed? Was it based on the number of base assignment differences between inferred and true clones?

10. Fig. 1h-1l: The presentation of the numbers of simulated mutations and clones at each panel will be useful.

Reviewer #2

(Remarks to the Author)

Shafiqhi et al. have developed a statistical method to map genomic clones to spatial data. Due to the highly clonal nature of tumors, authors have utilized paired whole exome sequencing to define tumor clones which are then mapped to the spatial data.

This tool is very timely in the community during a time when spatial genomic data is quickly growing and is a much needed statistical application of an important biological question relevant to pan-cancer data. While many teams are poised to analyze clonal transcriptional data, this group has attempted to integrate genomic alterations to inform clonality in this work.

While I think this work is very interesting and necessary I do have several major comments:

Line 177 – Are the 11461 cancerous spots identified across the 3 slides of the breast tumor? This seems to be extremely high for three ST slides given the authors initial indication that there are approximately 4885-4992 spots per sample. Does this indicate that all three slides are predominantly tumor dense with very little stroma? While pathology annotated slides are included in figure 3 – larger and much higher resolution images are required to justify such tumor dense regions without overlapping annotation to obscure the image.

Figure 3e – For clones in SB1 – Why is it that clone 2 and clone 6 annotate the same region of the H&E image when they are located in different regions along the tree? Is this due to coverage? Or overall nUMI counts per spot for various regions of the tumor section? Can you please provide some summary metrics for these three sections to show uniform UMI mapping across the sections to show there is no bias from technical artifacts? Is this just a very invasive tumor sample? I wonder are any DCIS regions annotated on this breast slide?

Line 199-200 “there was no detectable spatial pattern of domination of clones in sub-areas, as all clones were present in all sections uniformly” – While this is suggested by the data – this tends to go against the norm for previous breast cancer spatial studies – Here are a few publications I am referencing:

- <https://iopscience.iop.org/article/10.1088/1478-3975/abbe99> – STARCH for example uses copy number alterations to infer clonal structure on 4 layers of a breast cancer sample and show clones are not intermixed.
- <https://www.nature.com/articles/s41586-022-05023-2> - copy number based but tumor clonality still is not as intermixed.
- <https://www.nature.com/articles/s41586-022-05425-2> - using a new method to support the idea of subclone territories.

There are multiple places within the text where authors have mentioned custom scripts that are not accessible for reviewers:

1. In Methods – line 407-409 – authors mention a custom script to annotate pathologist data to ST data. This should be linked in a github page or other public repository to evaluate the tool.
2. Similarly in methods line 411-415 – authors mention another custom script that should be available to both reviewers and readers to interpret the feasibility to evaluate the task at hand. In particular for this step – who manually counted the cells? Were pathologists involved to validate cell calling in images?

The authors compare their tool to Cardelino. Since Cardelino also can provide an evolutionary tree based on bulk WES, it would be good to compare the generated tree from both models to confirm if the clonal structure is similar.

The authors allude to the copy number related tools that have been developed and utilized for clonal annotation to ST data. Given that they also have bulk WES data for these samples, can you further confirm the distinct clones by copy number variation and map to the ST data to validate the clonal structure. To date copy number has been the preferred clonal annotation for ST/singlecell RNAseq data due to a larger proportion of genes being covered by a given copy number region providing more evidence supporting a decrease or increase in expression over a larger area. Due to sparse coverage for point mutations and overall variability across the ST section, validation using an orthogonal method such as copy number can provide strong support for this mutation based model.

While, maybe out of the scope of this project. It would be interesting if the authors could look at breast cancer ST samples that have DCIS regions annotated. DCIS regions are enclosed by a myoepithelial layer and should theoretically be one clone in a given region.

Finally, this work would benefit from paired single-cell RNAseq of the same sections analyzed with bulk WES and ST data. While the authors are inferring clonal regions using WES based point mutation identification, there is no independent validation of tumor subclones from a transcriptional perspective. For example, for the annotated tumor region in the ST data – how many subclusters of tumor regions are identified by transcriptional signature? Of those related transcriptional signatures, how many of them have a mutation that could be mapped to that spot or site?

I have several suggestions and clarifications:

Line 115 – For coverage – are you referring to number of reads spanning a given site? Do these include split reads? Or only reads aligned with a specific mapping quality? These types of details should be accurately described in methods.

Line 132-134 – This is a very important statement that can inform suggested sequencing depth for future projects. Can you specify for a given sequencing depth what is the probability of capturing key genomic alterations to link to different tumor clones?

Figure 2 – boxplots should not be overlapping one another. Additionally, It would be beneficial to include background points that helped contribute to each distribution to give the reader a better idea about the changes by simulations. Finally, I am assuming for these boxplots that the triangles are outliers – but this is not noted in the legend.

For regions in the ST data that have high tumor infiltration how does this effect your classification? It would be important to note certain biological constraints on this method in the discussion and for future development.

I am interested to know how bulkDNA sequencing variant calling and coverage compared with ST data coverage. Can you provide a plot or follow up analysis to show how well covered different known driver mutations are represented in both datasets?

Line 433-434 – “The reads with a different nucleotide as compared to the reference genome were called the alternated reads.” For this statement are you just suggesting that any alternate call is acceptable? Why not take advantage of knowing the mutation to accurately call a true or false mutation as opposed to any different nucleotide from reference.

Line 595-595 – Why are different parallel chains used for the different cancer types – it is not obvious from the current description.

Line 140 – Annotations of Figures are out of order - Fig. 2e if mentioned before 2c.

Extended Data Figure 1 – In legend axes should be axes. And the annotation of high to very low read coverage seems to be inverted to the number of reads in each spot. Doesn't 2246 indicate high read coverage while very low should be 297 reads? Additionally to make the legend easier to access for the reader I would suggest copying and pasting the legend from Figure 2 here as well if parts of the figure legend overlap instead of referring back to an earlier figure legend. I am assuming for these boxplots that the triangles are outliers – but this is not noted in the legend.

Line 161 – nunumber -> number

Line 169 – Authors mentioned deep whole exome sequencing data was generated for breast samples but no methods are indicated relating to WES.

Line 173-174 – “We considered 608 high-confidence somatic single-nucleotide mutations (SNVs) identified from WES data that were co-observed in the annotated ST data (Methods)” – A description of this analysis is not found in methods as indicated.

Figure 3B – I recommend annotating a few mutations of interest on the heatmap to highlight known drivers or clones in the breast cancer samples. For certain clones – for example Clone 4 – how can the authors justify that there is not just a lack of coverage of certain genomic positions which differentiate clone 4 from 5-7?

Line 200 – underlined should be underscored.

Line 225 – “1-188 cells per spot” – It is a bit hard to believe that there could be 188 cells in an individual spot since on average using this technology the average cell count per spot is approximately 10 cells. Also the statement in line 303 suggesting the cells per spot count to be between 1-40 across both datasets seems to contradict the previous statement.

Figure 5 – Color bar legend is not annotated and neither are the colors associated with the clusters dendrogram. Gene names should also be italicized if discussing RNA-seq derived gene names.

Version 1:

Reviewer comments:

Reviewer #1

(Remarks to the Author)

The authors have made many revisions to improve the manuscript using my feedback. Therefore, I have no further comments.

Reviewer #2

(Remarks to the Author)

Major Comments:

In the reviewer response section the authors mention “We added a new section “Whole-exome sequencing for breast cancer samples” to Methods; page 22, paragraph 3. ” – but this section does not seem to be present. On line 198 – authors mention they performed deep whole exome sequencing but nothing is indicated in methods to suggest what is meant by “deep”. A coverage indication or methods on sequencing is required. While the authors cite other publications for bulk DNA and spatial sequencing methodology some abbreviated methods should be included in the methods of this work.

In addition to the breakdown of clones from canopy – it would be helpful to see – either in a table or figure – what percent of each clone is predicted by the WES. This would provide a stronger comparison between the percent distribution observed in ST and the ground truth from WES. Keeping in mind there will be some variation given that a different piece of tissue is used for bulk/ST.

I was very happy to see that a recent tool STARCH was incorporated into the new revised manuscript.

Did the authors evaluate if the breast cancer sample/prostate had copy number variations? This could help describe why STARCH didn't perform well on this particular sample. In such a case I don't think it is a fair comparison to think STARCH would perform better given there are no copy number alterations present in the sample. I understand that this is the benefit of this tool – but it would be helpful if copy number related data derived from WES could help support that copy number alterations are largely lacking in this dataset. This should be possible with the current WES and/or single cell data that the authors currently have.

STARCH methods and parameters are not currently included in the methods section. STARCH has the capability to identify multiple clones but if the correct input parameters are not used then that could explain why only two such clones are identified using the current run parameters. For example in the paragraph discussing STARCH usage on the breast cancer data – the authors mentioned they ranged the number of clones from 1-8 but only mentioned one clone is identified. Is only one clone observed even when the # of predicted clones parameter is increased? For example knowing the number of true clones based on WES – and then setting num_clusters = known value. Again this would be a good place to use the copy number data evaluated from the WES to help inform this parameter choice. I understand the authors mentioned they used the silhouette score – but I cannot find any description of utilizing this to set optimal number of clusters in the current STARCH methods.

While I appreciate the addition of the scRNA-seq data to validate the clusters – additional methods are required to evaluate how the authors funneled their cell count down to ~300 cells for analysis. Based on how much tumor content is present, I am surprised by the total tumor cell presented in this analysis. Further – could you also utilize mutation mapping on the single cell data to further validate your claims of the clusters for overlapping samples. I have found the current data presented in Extended Data Fig 21 to be quite confusing but think this could be a very strong finish to your current publication if corrected appropriately. Are all tumor cells from all three breast cancer samples shown in this figure? Is each point a cell?

Minor Comments:

Thank you for providing the higher resolution H&E images. I still think it is worth providing H&E images that are unmarked adjacent to the Annotated H&E images.

Line 68-70 - This sentence in introduction should be supported by a literature reference:
“For some cancers, only very few or no copy number alterations occur during disease progression.”

Line 104 - This statement seems redundant since alternated reads are reads containing mutations? If this is not the case please clarify. “alternated reads and the total number of reads for each mutation (mutation coverage)”. Maybe it should read as what is indicated in Line 110 instead?

Y-axis label for Figure 4c is missing.

Fig 4b – legend of color bar is not annotated.

Fig 4g – colored spots in the cardelino figure are not annotated.

Figure 3 caption – presentation is misspelled.

Extended Data Figure 6 – Having a few more values on the y axis will be helpful for interpretation. Further if the spots could include common variant/other variant annotation.

In extended data fig 9 – an annotation is needed for the color legend. Is it showing the number of reads that are alternated/total etc? Why is the read count less than 1? Or is it proportion of reads?

COSMIC should be capitalized in Fig. 3 caption.

I appreciate the addition of Extended Data Figures 8-9. For extended data fig 9 please modify the color scheme as it is very

hard to interpret the read coverage given the outliers really take over the entire plot.

I appreciate the time put into making the code accessible and available to support the work presented in this study.

Version 2:

Reviewer comments:

Reviewer #2

(Remarks to the Author)

Only one minor comment!

On extended data figure 11 a – “cardelinoo average” should be “cardelino average”

We are grateful to all Reviewers' comments. We believe that Reviewers' suggestions and the consequent revision have greatly benefited the manuscript. Below are detailed answers (marked in blue) to each comment.

ANSWERS TO REVIEWER COMMENTS

Reviewer #1 (Remarks to the Author):

The authors developed a new method, Tumoroscope, to map inferred clones in the tumor tissue for spatial transcriptomic (ST) data. Using bulk DNA sequencing data, it first infers clones by using an available clone prediction method, Canopy. Then it uses read counts from different spots (mini-bulk sequences) in the tissue to estimate the frequency of each clone at each spot. The approaches used are standard, but the application domain is important. There is relatively little by the way of innovation, even though I see room for much innovation here.

We strongly disagree with the claim that the approaches are standard and that there is little innovation in our approach. In fact, we propose a highly innovative probabilistic graphical model, with unprecedented functionality. There exists no comparable approach for the same task of mapping clones to tissues using single nucleotide variants and inferring their gene expression profiles. The model is also sophisticated methodologically. The full depth of the Tumoroscope model can be appreciated in the Methods section. To improve the presentation of the novelty of the approach, we modified **page 3, paragraph 3**.

The comparison with Cardelino using simulated datasets, which is not really designed for spatial transcriptomic data, is presented. This is not a very informative comparison because it is designed to map inferred clones to single cells using SNVs from single-cell RNA-seq data.

Thank you for this comment. We carefully and deliberately selected Cardelino as an important alternative approach to compare to. This comparison brings a highly relevant piece of information: it shows how much the heterogeneity inside each spot in terms of the mixture of cell types matters. If spots were homogeneous, Cardelino would be perfectly applicable to the task of mapping clones in tissues, since each spot would correspond to a single cell, with even a higher coverage than single cells usually have, and thus more statistical power for mapping with Cardelino.

The comparison shows that thanks to the ability of Tumoroscope to deconvolute clonal composition in each spot, its performance is much higher. This point is now clarified and the comparison to Cardelino as a very important aspect is motivated in **page 7 paragraph 3**. Note since there is no other computational approach with the same functionality as Tumoroscope, we cannot compare it to any other existing method on exactly the same task. In this revision we enlarged the set of alternative compared approaches and we also applied STARCH - a method that maps clones in tissue space, but based on copy number profiles (and not on somatic point mutations, as Tumoroscope does).

1. Clone decomposition from a single bulk sample is extremely difficult empirically and theoretically. One cannot believe that the Canopy (the method used) can accurately infer clones, and the impact of errors in the clone prediction is expected to be large, which is a major weakness of this work.

Thank you for this comment. This concern is indeed valid, and we acknowledge it in our study. However, it is important to note that we employed three distinct samples from the same patient, rather than relying on a single bulk sample, which enhances the reliability of our results.

While it is true that various models, such as Canopy and others, exhibit varying levels of accuracy, they still provide valuable tools for tumor evolution inference. Despite potential doubts about their individual accuracies, these models represent the current state-of-the-art toolbox available for this purpose, and as researchers, it is essential to utilize the best resources at our disposal. As they are improved, our framework can take advantage of the newer better version.

In our study, we recognize that one potential weakness lies in the dependence on the correctness of Canopy (or any other method used for tumor evolution inference as a preprocessing step), which we have addressed in the **Discussion section (page 21, paragraph 2)**. Moreover, we highlight that our proposed model is flexible and can accommodate other, potentially more accurate tools as they become available. If a more precise method for tumor evolution inference emerges in the future, we can readily integrate it into our approach to potentially achieve even better results and deepen our understanding of tumor development.

2. It is unclear how Tumoroscope use the information on CNAs? In Tumoroscope, Canopy is performed by giving the information of CNAs. I guess inferred clones from Canopy include the status of CNAs?

Indeed, the information about CNAs is accounted for by Canopy as well as by our model, Tumoroscope. The input to Canopy is the inferred copy number for the bulk sample (we infer it using FalconX) and Canopy is using this information to infer more accurate SNV clones. The genotypes inferred by Canopy represent binary values. We modify these genotypes using the allele specific copy number of mutated alleles relative to the total copy number. This allele-specific copy number information is fed to Tumoroscope. This additional copy number insight benefits our model, as it relies on matching mutations between the genotypes and the variants observed in ST data. The absolute copy number itself becomes irrelevant, and instead, the relative abundance of mutated alleles holds significance.

We highly appreciate your insightful comment, and as a result, we have incorporated this vital piece of information into **Figure 1** and expanded the model's description in the **Results section (page 4, paragraph 1)**.

3. Clone frequency at each spot is estimated using observed read count within a spot. However, the read counts are obtained from RNA-seq. So, over-expression and under-expression of some genes at a spot will affect the observed read count. Also, some mutant alleles are not expressed (Househam et al., Nature. 2022. doi: 10.1038/s41586-022-05311-x). How will all this affect the performance of clone fraction (frequency) estimation in the Tumoroscope?

Thank you for your valuable comment, which highlights a major strength of our model that may not have been adequately explained. Tumoroscope leverages the observed counts of alternative and total reads, utilizing the Binomial distribution to compute the model likelihood—this critical information forms the primary signal for our data analysis. An essential advantage of this approach is its independence from gene expression fluctuations, as the ratio of alternative to total reads remains unaffected. We have clarified this aspect in greater detail in **page 4, paragraph 1** and **page 27, paragraph 3**.

Regarding non-expressed mutant alleles, we acknowledge that ST may not capture all mutations, but it is not necessary to capture every single mutation for our model's purpose. It is sufficient when a subset of mutations is available to match their signals to the corresponding clones, and when the subset of expressed mutations ensures significant mutational differences between the clones. To additionally validate the robustness of our model and its ability to handle missing data, based on your suggestion we simulated scenarios with 0%, 25%, 50%, and 75% of missing mutations. These simulations confirmed our model's accuracy in reconstructing clone fractions, providing us with greater confidence in its reliability (see **Extended Data Figure 3**).

We sincerely appreciate your comment, which significantly contributed to the comprehensive analysis and validation of our model. Your feedback has strengthened our study, and we are pleased to include these additional essential findings to improve the overall quality of our research.

4. Other methods for spatial transcriptomic data use CNAs to identify clones, e.g., STARCH (lines 333-337). It seems CNAs are common in cancer. When data is affected by CNAs, will Tumoroscope perform better than the existing methods?

Indeed other methods are based on CNAs only. If the tumor evolution is driven solely by CNVs, they can successfully be applied to resolve spatial composition of the clones in ST data. Arguably, SNVs have been even more comprehensively studied as evolutionary events than CNVs, and their functional roles are better understood across cancer types. Moreover, it may be the case that no substantial CNV changes occur in tumor subclones and CNV-based methods such as STARCH are not applicable for identifying spatial genomic heterogeneity. Therefore, analyzing SNV-based clones provides critical and complementary insights into cancer

progression that should be prioritized rather than overlooked. This important argument is now stated **in the Introduction page 3, paragraph 1.**

It is important to note that STARCH and Tumoroscope offer distinct interpretations of tumor clones—either through shared copy number changes or shared point mutations. This divergence implies that methods such as STARCH might overlook clones identified by SNVs that Tumoroscope can pinpoint, and vice versa, rendering the comparison of these two approaches difficult.

Still, we did approach the comparison of results of Tumoroscope to STARCH (see **Results,, page 12, paragraph 5; Results, age 13, paragraph 1; Results, age 15, paragraph 7; Results, page 16, paragraph 1; Discussion, age 20, paragraph 3; Discussion, page 21, paragraph 1; Fig. 4; Methods, age 32, paragraphs 2 and 3; Fig. 6**). To this end, we ran STARCH for 1 to 8 clusters (beside the normal clone) and applied the default recommended way of choosing the number of clones in STARCH (using the Silhouette score; see **Extended Data Figure 12**). Both for prostate and breast cancer samples STARCH identified only one CNA tumor cluster alongside with the normal clone. This demonstrates that although both tumor samples showed substantial heterogeneity in SNV-based clones, they are not heterogeneous in terms of copy number-based clones, which is supported by the results of FalconX on WES. The inability to resolve clones by STARCH for this data is further shown in the cancerous prostate tissue region, where STARCH indicated almost half of the spots as normal while H&E data clearly indicated that a vast majority of cells in this region are cancerous, and Tumoroscope succeeded to find three tumor clones for prostate and six for breast cancer (**See Figure 3,4,5 and 6**). The reason for STARCH marking cancer tissue as normal might be that some true subclones evolved only via SNV events and for those STARCH cannot identify any tumor clone at all. The reason for STARCH identifying only a single tumor clone for the other part of the cancerous tissue might be that for the remaining subclones the copy number events occurred early in tumor evolution and are present in all cells in those subclones and thus do not display any spatial CNV heterogeneity; in those subclones, SNVs happened later in the evolutionary process and led to the formation of the clones. This shows that we definitely need the Tumoroscope model to analyze the tumors where the copy number-based models can not even identify the tissues as cancerous and cannot identify subregions with distinct clones.

5. Also, was the empirical data affected by CNAs? If so, it needs to be discussed to interpret the results of the Tumoroscope, e.g., gene expression pattern for those affected by CNAs.

Thank you for this comment, again enriching our manuscript. As discussed in detail in the previous answer, both analyzed cancer tissues (prostate and breast) were only little affected by CNVs (as determined both by FalconX and STARCH). To investigate whether Tumoroscope is negatively affected by copy number events, we separated the genes based on their copy number state (0,1,2) as inferred by STARCH and showed the expression of the genes inferred

by Tumoroscope in each copy number state (see **Extended Data Figure 17, 18 and 19; Results: page 18, paragraph 2**).

This analysis demonstrated that for our data the expression levels of genes inferred by Tumoroscope agree with copy number levels inferred by STARCH very well. This result further confirms that our model is not adversely affected by the occurrence of CNVs and can correctly infer clone-specific expression profiles irrespective of their copy number. However, this very good agreement may be a result of the particularly low contribution of CNV events to the evolutionary process behind the analyzed prostate and breast cancer data. We hypothesized that if there were more CNV tumor clones than only one, and they would split SNV-based clones in evolution, the agreement would be less pronounced.

6. Scripts and settings to process ST and bulk data for Tumoroscope analysis should be available for others. Also, demonstration of some example data analysis starting from processing raw ST data will be useful for others to use Tumoroscope for their data.

Based on your helpful feedback, we have expanded the repository with two new folders. One is the Vignette folder containing a static HTML page and interactive Jupyter notebook that walks through using Tumoroscope end-to-end. This new vignette aims to make the workflow accessible and easy to follow for new users. Both versions can be found in the new Vignette folder on the model repository (<https://github.com/szczurek-lab/Tumoroscope>). The other folder is a Demo which contains the demo code for the simulation and validation of the model.

We also published our repository for the read counting script (https://github.com/szczurek-lab/st_read_counter) and added this information to the “**Code availability**” section of the paper.

7. An extensive simulation study was performed, so simulated data should be available. Importantly, biological justification for the selection of parameters used in the computer simulation is needed, particularly because the biological datasets available are rather limited at present.

Thank you for your valuable feedback. A comparison between the parameters of the actual data and the simulated data is presented in **Extended Data Table 1**. The simulated data is now available at <http://doi.org/10.5281/zenodo.10255434> and this information is added to the “**Data availability**”. Besides, each simulated dataset can be independently sampled using our code in a reproducible manner. For this purpose we have included a demo code in the GitHub repository, enabling users to effortlessly generate datasets with their desired parameters using a config file ([Tumoroscope/demo at main · szczurek-lab/Tumoroscope \(github.com\)](https://github.com/szczurek-lab/Tumoroscope/blob/main/Tumoroscope/demo)).

8. How was a mean average error (MAE) computed? MAE was defined as “the average of the difference between the inferred proportions of the clone and the true values in all the spots and clones.” Specifically, how was inferred clone annotated to the true clone?

Tumoroscope is capable of detecting the portion of a clone within a spot, utilizing the clone's genotype. In our approach, we possess both a vector representing the actual clone fractions for each spot and a vector denoting the clone fractions inferred by Tumoroscope for the same spots. To compute the error, we calculate the average difference between these two vectors, providing us with the Mean Average Error (MAE) value for that specific spot. This process is repeated for all spots, and the average MAE values across all spots are derived. For a more detailed explanation, refer to **paragraph 1 on page 6**.

9. Fig. 1h-1l and lines 151-153: How was the accuracy computed? Was it based on the number of base assignment differences between inferred and true clones?

To ensure a fair comparison to cardelino, we exclusively examined the major clone identified by Tumoroscope within each spot (major meaning the clone with the highest inferred fraction for that spot). We quantified accuracy as the percentage of agreement between the major inferred clone and the true major clone in our simulated data. This explanation is outlined in **paragraph 3 on page 7**.

10. Fig. 1h-1l: The presentation of the numbers of simulated mutations and clones at each panel will be useful.

Thank you for your suggestion. All simulated data presented in **Figure 2** featured 5 clones and 30 mutations. This essential detail has been incorporated into the caption of **Figure 2**.

Reviewer #2 (Remarks to the Author):

Shafiqhi et al. have developed a statistical method to map genomic clones to spatial data. Due to the highly clonal nature of tumors, authors have utilized paired whole exome sequencing to define tumor clones which are then mapped to the spatial data.

This tool is very timely in the community during a time when spatial genomic data is quickly growing and is a much needed statistical application of an important biological question relevant to pan-cancer data. While many teams are poised to analyze clonal transcriptional data, this group has attempted to integrate genomic alterations to inform clonality in this work.

While I think this work is very interesting and necessary I do have several major comments:

Line 177 – Are the 11461 cancerous spots identified across the 3 slides of the breast tumor? This seems to be extremely high for three ST slides given the authors initial indication that there are approximately 4885-4992 spots per sample. Does this indicate that all three slides are predominantly tumor dense with very little stroma? While pathology annotated slides are included in figure 3 – larger and much higher resolution images are required to justify such tumor dense regions without overlapping annotation to obscure the image.

According to the pathologist's annotations, a total of 11,461 cancerous spots were identified across six slides (not three slides, there are two nearby samples per each of the three locations). In Figure 3e, we showed three of these slides for clarity, noting that the annotations for two adjacent slides in each section were identical. The figure highlights the prevalence of cancerous spots in sections SB1 and SB2, where they are particularly dominant.

Additionally, we added high-resolution figures as additional files reposted at <http://doi.org/10.5281/zenodo.10255434> (now linked in the **Data availability** section).

Figure 3e – For clones in SB1 – Why is it that clone 2 and clone 6 annotate the same region of the H&E image when they are located in different regions along the tree? Is this due to coverage? Or overall nUMI counts per spot for various regions of the tumor section?

Thank you for your insightful comment. Indeed, we agree that colocalization of clones 2 and 6, which are relatively far away in the tree, is an interesting and surprising finding. The clones 2 and 6 differ by multiple mutations (their pattern can be appreciated in **Figure 4a**).

By construction, the model locates clones in spots by matching mutations that are characteristic of clonal genotypes. Of course, the mapping is not deterministic, but only probabilistic, as the occurrence of mutations in spatial transcriptomics is highly noisy and obscured by missing data (as the coverage is low). Still, the inherent logic of the model indicates that clones 2 and 6 can only be identified by Tumoroscope as colocalized when

mutations specific for clone 2 and not present in clone 6 and vice versa (specific for clone 6 and not present in clone 2) are predominantly found in the same areas.

To validate this observation, we used the alternate read counts corresponding to mutations found in clone 2 but not in clone 6, and conversely, mutations present in clone 6 but absent in clone 2 visualized in new **Extended Figure 8a**). This analysis provides compelling confirmation of colocalization of the alternated reads for the mutations specific to clone 2 and specific to 6 within the specific region where Tumoroscope has also identified these clones.

In addition we plotted both the alternated and total reads in each spatial spot (new **Extended Figure 9**). Interestingly, the colocalization of clones 2 and 6 also coincides with higher coverage areas. This however, does not affect the identification of the clone location in Tumoroscope. Instead, the higher coverage should be accounted for in the model by inferring higher variant expression values (Φ variables in the model) for the variants in these clones (confirmed in new **Extended Figure 10**).

These new investigations are now described in **page 11, paragraph 3 and page 12, paragraph 1**.

Can you please provide some summary metrics for these three sections to show uniform UMI mapping across the sections to show there is no bias from technical artifacts? Is this just a very invasive tumor sample? I wonder are any DCIS regions annotated on this breast slide?

Thank you once more for your insightful comment. In response to your suggestion, we have incorporated additional visualizations into our analysis. Specifically, we have included plots illustrating alternated and total read counts for all sections of prostate and breast cancer data (new **Extended Figure 9**). While it is expected that coverage varies due to technical artifacts and the activity of different cancer clones, our findings demonstrate the presence of mutated reads across all tissues, affirming their cancerous nature. Additionally, consistent with the evaluation by our pathologist, these samples have been classified as invasive tumor specimens without any isolated DCIS regions.

Line 199-200 “there was no detectable spatial pattern of domination of clones in sub-areas, as all clones were present in all sections uniformly” – While this is suggested by the data – this tends to go against the norm for previous breast cancer spatial studies – Here are a few publications I am referencing:

- <https://iopscience.iop.org/article/10.1088/1478-3975/abbe99> – STARCH for example uses copy number alterations to infer clonal structure on 4 layers of a breast cancer sample and show clones are not intermixed.
- <https://www.nature.com/articles/s41586-022-05023-2> - copy number based but tumor clonality still is not as intermixed.

- <https://www.nature.com/articles/s41586-022-05425-2> - using a new method to support the idea of subclone territories.

Please note that the outcomes that you mentioned relate to the intermixed clones found by cardelino and not Tumoroscope. In contrast, the results of Tumoroscope show remarkable sub-localization of the clones, even though we do not utilize spot position information as STARCH does. Therefore, the results of our approach (Tumoroscope) agree with the previous references that you mentioned, while for cardelino they do not.

There are multiple places within the text where authors have mentioned custom scripts that are not accessible for reviewers:

1. In Methods – line 407-409 – authors mention a custom script to annotate pathologist data to ST data. This should be linked in a github page or other public repository to evaluate the tool.

Thank you for pointing this out. We added the GitHub repository ([szczurek-lab/qupath-spot-utils: Scripts for processing microarray spots in QuPath \(github.com\)](https://github.com/szczurek-lab/qupath-spot-utils)) of the script to align H&E image data to ST spots and counting the cells to the “**Code availability**” section.

Additionally, in the “**Data availability**” section we provided links to all data sources, so that both previously and newly published data are easily accessible.

2. Similarly in methods line 411-415 – authors mention another custom script that should be available to both reviewers and readers to interpret the feasibility to evaluate the task at hand. In particular for this step – who manually counted the cells? Were pathologists involved to validate cell calling in images?

We made the script openly available here:

[szczurek-lab/qupath-spot-utils: Scripts for processing microarray spots in QuPath \(github.com\)](https://github.com/szczurek-lab/qupath-spot-utils) and added it to the “**Code availability**” section.

The manual counting was performed using our experienced H&E image analyst (Igor Filipiuk), in close collaboration with the pathologist Łukasz Koperski.

The authors compare their tool to Cardelino. Since Cardelino also can provide an evolutionary tree based on bulk WES, it would be good to compare the generated tree from both models to confirm if the clonal structure is similar.

Cardelino is incapable of generating a tree independently; it relies on the tree generated by canopy as its input. Notably, we employed the same tree for both Tumoroscope and cardelino in our analysis, and so both Tumoroscope and cardelino were working on the exact same trees.

The authors allude to the copy number related tools that have been developed and utilized for clonal annotation to ST data. Given that they also have bulk WES data for these samples, can

you further confirm the distinct clones by copy number variation and map to the ST data to validate the clonal structure. To date copy number has been the preferred clonal annotation for ST/single cell RNAseq data due to a larger proportion of genes being covered by a given copy number region providing more evidence supporting a decrease or increase in expression over a larger area. Due to sparse coverage for point mutations and overall variability across the ST section, validation using an orthogonal method such as copy number can provide strong support for this mutation based model.

Indeed, the identification of mutations from transcriptomics data poses a significant challenge. Our method effectively addresses this challenge using probabilistic modeling, enabling us to harness additional datasets such as whole-exome sequencing (WES) and H&E staining to supplement the evidence gap.

Copy number variant (CNV) evolution and single nucleotide variant (SNV) evolution have traditionally been investigated as separate entities, with distinct computational tools tailored for each type of genetic alteration. Consequently, these methods offer distinct insights into tumor clones, which are either defined by shared copy number changes or shared point mutations. This divergence implies that STARCH might miss clones identified by SNVs, which Tumoroscope can accurately pinpoint, and vice versa.

With this context in mind, we conducted a comprehensive comparison of the spatial arrangement of the SNV-clones found using Tumoroscope and CNV-clones found using STARCH (see **Results, page 12, paragraph 5; Results, page 13, paragraph 1; Results, page 15, paragraph 7; Results, page 16, paragraph 1; Discussion, page 20, paragraph 3; Discussion, page 21, paragraph 1; Fig. 4; Methods, page 32, paragraph 2 and 3; Fig. 6**). To identify the number of clones for STARCH, we followed the default recommended method, which involves using the Silhouette score (see **Extended Data Figure 12**).

For both prostate and breast cancer samples, STARCH identified only one CNA tumor cluster alongside the normal clone. This highlights a crucial observation: although both tumor samples exhibited substantial heterogeneity in SNV-based clones found using Tumoroscope, they lacked heterogeneity in terms of copy number-based clones.

The limitations of STARCH become more evident when examining the cancerous prostate tissue region. STARCH categorized almost half of the spots as normal, despite H&E data clearly indicating that the cells in this region are cancerous. In contrast, Tumoroscope successfully identified tumor clones for prostate cancer in these spots (see **Figure 6**).

The reason for STARCH incorrectly classifying cancerous tissue as normal might be that certain true subclones evolved solely through SNV events, making them undetectable by STARCH. STARCH's identification of only a single tumor clone in other parts of cancerous tissue could

be due to copy number events occurring early in tumor evolution, affecting all cells in those subclones and thus lacking spatial CNV heterogeneity. In contrast, SNVs could have occurred later in the evolutionary process, leading to the formation of distinct clones.

These findings underscore the significance of the Tumoroscope model in analyzing tumors where copy number-based models struggle to identify cancerous tissues and distinguish subregions with unique clones (**Discussion, page 20, paragraph 3**).

While, maybe out of the scope of this project. It would be interesting if the authors could look at breast cancer ST samples that have DCIS regions annotated. DCIS regions are enclosed by a myoepithelial layer and should theoretically be one clone in a given region.

As per our pathologist's (Łukasz Koperski) evaluation, confirmed by a consilium of additional pathology experts, these samples have been categorized as invasive tumor specimens, without any isolated DCIS regions.

Finally, this work would benefit from paired single-cell RNAseq of the same sections analyzed with bulk WES and ST data. While the authors are inferring clonal regions using WES based point mutation identification, there is no independent validation of tumor subclones from a transcriptional perspective. For example, for the annotated tumor region in the ST data – how many subclusters of tumor regions are identified by transcriptional signature? Of those related transcriptional signatures, how many of them have a mutation that could be mapped to that spot or site?

We appreciate your helpful feedback, which led to an important improvement in our study. We investigated additional RNA-seq data from individual cells from the same breast cancer patient and tissue sections. We first independently clustered the single cells based on their gene expression profiles and next mapped the gene expression profiles of the clones inferred by Tumoroscope to the closest cluster. Interestingly, we found that clone 2 and 6, as well as clone 3 and 5 that are co-localized together in the ST spots as inferred by Tumoroscope, mapped to the same gene expression clusters based on the orthogonal analysis by single cell data. These confirmatory results are now described in a new subsection of the Results and visualized in two new Figures (**Extended data Figure 20 and 21 ; page 18, paragraph 3; page 19, paragraph 1 and 2**).

I have several suggestions and clarifications:

Line 115 – For coverage – are you referring to number of reads spanning a given site? Do these include split reads? Or only reads aligned with a specific mapping quality? These types of details should be accurately described in methods.

For ST data, we use the terms spot coverage and mutation coverage to refer to the number of reads found in a spot or over a mutation, respectively.

This is now clarified in the text (**page 25, paragraph 2**). For real data, to determine the number of reads per variant site, we employ two main steps to process the data. First, the ST pipeline, consisting of quality checks, artifact removal, and genome alignment, guarantees data integrity (SpatialTranscriptomicsResearch/st_pipeline: ST Pipeline contains the tools and scripts needed to process and analyze the raw files generated with the Spatial Transcriptomics method in FASTQ format. (github.com)). Then, using our custom script, we calculate the number of remaining high-quality reads as coverage. This two-step approach ensures robust and accurate results. We have included this information in the **Methods** section, specifically in **paragraphs 2 and 4 on page 24**.

Line 132-134 – This is a very important statement that can inform suggested sequencing depth for future projects. Can you specify for a given sequencing depth what is the probability of capturing key genomic alterations to link to different tumor clones?

We appreciate your valuable feedback. In our simulation study, we conducted simulations and model comparisons across four different sequencing depths per variant per spot: 18, 50, 80, and 110, corresponding to an average per variant per cell coverage of 1, 3, 4, 5. As an additional result, we have now included visualizations that depict the probabilities of observing at least 'K' alternative reads per variant per cell in a spot in each of these scenarios, for K ranging from 1 to the total analyzed depth. These probabilities were calculated using the cumulative binomial distribution, with simplifying assumptions of a biallelic DNA and heterogeneous mutations, which yield a success rate of $\theta=1/2$ (**Results; paragraph 2, page 6; Extended Data Figure 1**).

Figure 2 – boxplots should not be overlapping one another. Additionally, It would be beneficial to include background points that helped contribute to each distribution to give the reader a better idea about the changes by simulations. Finally, I am assuming for these boxplots that the triangles are outliers – but this is not noted in the legend.

Thank you for your valuable feedback. We have addressed these concerns by ensuring that the boxplots do not overlap and adding the background in both **Figure 2** and **Extended data Figure 2**. Finally, we made sure the outliers are explained in the caption of each Figure.

For regions in the ST data that have high tumor infiltration how does this effect your classification? It would be important to note certain biological constraints on this method in the discussion and for future development.

We recognize that the presence of other cell types in highly infiltrated regions can be challenging for our model. In our theoretical framework, these cell types are anticipated to be classified as part of the normal clone due to their lack of specific tumor-associated mutations. We've incorporated this consideration into **Discussion** in **paragraph 2 on page 21**.

I am interested to know how bulkDNA sequencing variant calling and coverage compared with ST data coverage. Can you provide a plot or follow up analysis to show how well covered different known driver mutations are represented in both datasets?

We performed an analysis to identify mutations within known driver genes, specifically focusing on mutations that were shared between the ST and WES datasets. In the breast cancer dataset, we discovered 46 such mutations, and in the prostate cancer dataset, we found 18.

Subsequently, we created **Extended data Figure 6** to visualize the distribution of alternate and total read counts per cell per variant position in both the WES and ST data. The much larger coverage of mutations in WES as compared to ST underlines the importance of a probabilistic treatment of the problem of deconvolving spots into clones, as it is performed by Tumoroscope. We added this information in **Results; page 10, paragraph 2; page 13, paragraph 2; page 15, paragraph 1**.

Line 433-434 – “The reads with a different nucleotide as compared to the reference genome were called the alternated reads.” For this statement are you just suggesting that any alternate call is acceptable? Why not take advantage of knowing the mutation to accurately call a true or false mutation as opposed to any different nucleotide from reference.

Thank you for your comment. Not any alternate call is acceptable. We utilized nucleotide information from the WES data to select those alterations for which the specific alternative nucleotide matched consistently between the WES and ST data. We have now included this information in **Methods; age 24, paragraph 4**.

Line 595-595 – Why are different parallel chains used for the different cancer types – it is not obvious from the current description.

As explained in ‘**Clonal assignment of the spots using cardelino**’ section in Methods (**page X, paragraph X**) running multiple chains for datasets with large numbers of spots is prohibited by the significant RAM requirements of cardelino. The RAM demand grows with the number of spots. With 294 spots in the prostate datasets and 11,461 spots in the breast cancer dataset, it is nearly 39 times larger and becomes computationally infeasible. Such a large amount of RAM is not available at any of the computational servers that we have access to.

Line 140 – Annotations of Figures are out of order - Fig. 2e if mentioned before 2c.

Please kindly notice that a-f are mentioned together before mentioning one by one (**page 7 line 151**).

Extended Data Figure 1 – In legend axes should be axes. And the annotation of high to very low read coverage seems to be inverted to the number of reads in each spot. Doesn't 2246 indicate high read coverage while very low should be 297 reads? Additionally to make the legend easier

to access for the reader I would suggest copying and pasting the legend from Figure 2 here as well if parts of the figure legend overlap instead of referring back to an earlier figure legend. I am assuming for these boxplots that the triangles are outliers – but this is not noted in the legend.

Thank you so much for your comments. We improved the Figure as you suggested.

Line 161 – number -> number

We improved the text accordingly.

Line 169 – Authors mentioned deep whole exome sequencing data was generated for breast samples but no methods are indicated relating to WES.

We added a new section “Whole-exome sequencing for breast cancer samples” to **Methods; page 22, paragraph 3.**

Line 173-174 – “We considered 608 high-confidence somatic single-nucleotide mutations (SNVs) identified from WES data that were co-observed in the annotated ST data (Methods)” – A description of this analysis is not found in methods as indicated.

The relevant description can be found in the 'Bulk DNA-seq and somatic mutation calling' and 'Selection of somatic mutations detected in both bulk DNA-seq and ST data' sections of the Methods.

Figure 3B – I recommend annotating a few mutations of interest on the heatmap to highlight known drivers or clones in the breast cancer samples. For certain clones – for example Clone 4 – how can the authors justify that there is not just a lack of coverage of certain genomic positions which differentiate clone 4 from 5-7?

As you suggested, we identified and plotted the driver genes, cancer hallmarks, and the top 20 genes for each cancer among the clones in new **Figures 3 and 5.**

Additionally, we visualized the alternate read counts for mutations that are present exclusively in specific clones, focusing on the pairs of clones which we identified as spatially co-occurring (2 and 6, 3 and 5) or mutually exclusive (4 and 7; **Extended Figure 8**) . While Tumoroscope leverages both exclusive and common mutations simultaneously, the presence of clone-specific mutations confirms that the data provides signal needed for inference to the model.

Line 200 – underlined should be underscored.

This phrase was corrected.

Line 225 – “1-188 cells per spot” – It is a bit hard to believe that there could be 188 cells in an individual spot since on average using this technology the average cell count per spot is approximately 10 cells. Also the statement in line 303 suggesting the cells per spot count to be between 1-40 across both datasets seems to contradict the previous statement.

The 1-40 cell count estimates were suggested by the authors of the respective papers. However, the typical range of these data is larger, and we reformulated the range to 1-100 cells. In the Introduction we also explain that the actual observed values can vary between individual spots due to different technical parameters and cell sizes (**page 2, third paragraph**). To be able to precisely estimate the number of cells in each spot for our particular tissue samples, we developed a script in QuPath for counting the number of cells in each spot from H&E data, and the range 1-188 is based on the result of that algorithm. Such counting is prone to errors due to technical artifacts such as overlapping nuclei of nearby cells. An expert did inspect and confirm the validity of the results of this counting algorithm by eye. However, manually validating the number of all cells in all spots is impractical to perform by a human. Consequently, we adjust the inferred number of cells within Tumoroscope based on other input variables.

Thanks to your feedback, we have also visualized the number of cells counted using our script in QuPath alongside the numbers inferred by Tumoroscope (**see Extended Figure 7**). This comparison shows that Tumoroscope tends to reduce the numbers for spots with very high estimates and increase them for those with very low estimates. This behavior suggests that Tumoroscope is effectively correcting outliers, which may represent errors in the estimates (**Results; page 11, paragraph 1 and page 13 last paragraph**).

Figure 5 – Color bar legend is not annotated and neither are the colors associated with the clusters dendrogram. Gene names should also be italicized if discussing RNA-seq derived gene names.

Thank you for your comment. We have added annotations to the legend and italicized the gene names. The colors in the legend represent the intensity of gene expression inferred by Tumoroscope. The gene clusters are not discussed in the paper and were removed in the updated version of this Figure.

Thank you very much for your in depth inspection of the manuscript. We feel that after incorporating the changes that you have suggested, our paper has largely gained on quality. Below we address each comment, answering in blue.

Major Comments:

In the reviewer response section the authors mention “We added a new section “Whole-exome sequencing for breast cancer samples” to Methods; page 22, paragraph 3. ” – but this section does not seem to be present. On line 198 – authors mention they performed deep whole exome sequencing but nothing is indicated in methods to suggest what is meant by “deep”. A coverage indication or methods on sequencing is required. While the authors cite other publications for bulk DNA and spatial sequencing methodology some abbreviated methods should be included in the methods of this work.

We apologize for this confusion. To clarify the description of the newly generated and previously published data and to make it more self-contained, we have now divided the text into three Methods sections: “**ST experiments for breast cancer samples**”, “**Whole-exome sequencing for breast cancer samples**”, and “**ScRNA-seq experiments for breast cancer samples**” (pages 22, 23, and 23, respectively). We have specified the depth of whole-exome sequencing as 300x, which is now mentioned on **page 23, line 535**.

In addition to the breakdown of clones from canopy – it would be helpful to see – either in a table or figure – what percent of each clone is predicted by the WES. This would provide a stronger comparison between the percent distribution observed in ST and the ground truth from WES. Keeping in mind there will be some variation given that a different piece of tissue is used for bulk/ST.

Thank you for this comment. The information about the proportion of each clone in each section seen both in WES and in ST is provided in **Extended data Figure 11**.

I was very happy to see that a recent tool STARCH was incorporated into the new revised manuscript.

Did the authors evaluate if the breast cancer sample/prostate had copy number variations? This could help describe why STARCH didn’t perform well on this particular sample. In such a case I don’t think it is a fair comparison to think STARCH would perform better given there are no copy number alterations present in the sample. I understand that this is the benefit of this tool – but it would be helpful if copy number related data derived from WES could help support that copy number alterations are largely lacking in this dataset. This should be possible with the current WES and/or single cell data that the authors currently have.

We did find the copy number variation on the level of bulk samples for both cancer types, using FalconX. This finding aligns with the result of STARCH, which, in addition to the healthy clone, finds a cancer clone containing copy number variation.

However, it is essential to note that whole exome sequencing (WES) does not allow us to identify individual copy number clones, their quantities, or their specific copy number variants. WES provides information on copy number variations only in an aggregate manner. Consequently, WES analysis cannot directly support or refute the findings of STARCH.

STARCH methods and parameters are not currently included in the methods section. STARCH has the capability to identify multiple clones but if the correct input parameters are not used then that could explain why only two such clones are identified using the current run parameters.

Thank you for this comment. To clarify the way we used STARCH in this analysis, we added a new Methods section “**Assigning spots to copy number clones using STARCH**” (page 32).

We would like to emphasize that the run command for the STARCH model was based on the authors' GitHub page: [raphael-group/STARCH: Spatial Transcriptomics Algorithm Reconstructing Copy-number Heterogeneity \(github.com\)](https://github.com/raphael-group/STARCH: Spatial Transcriptomics Algorithm Reconstructing Copy-number Heterogeneity).

The only parameter that needs adjustment when running STARCH is the number of clusters (clones), denoted as K . According to the authors' supplement file, Section 6: Model Initialization, page 4, they recommend: "The number K of clones may be selected either using prior knowledge or by computing the average silhouette score for a range of K and selecting the value of K with the highest average silhouette score." This recommendation guided our procedure for determining the appropriate number of clusters.

For example in the paragraph discussing STARCH usage on the breast cancer data – the authors mentioned they ranged the number of clones from 1-8 but only mentioned one clone is identified. Is only one clone observed even when the # of predicted clones parameter is increased? For example knowing the number of true clones based on WES – and then setting `num_clusters = known value`. Again this would be a good place to use the copy number data evaluated from the WES to help inform this parameter choice. I understand the authors mentioned they used the silhouette score – but I cannot find any description of utilizing this to set an optimal number of clusters in the current STARCH methods.

Following the Reviewer's suggestion, we evaluated the number of spots assigned by STARCH to different clones, increasing the number of cancer clones up to 8 for both breast and cancer samples (as shown below). However, it is important to note that despite the increase in the number of clones with higher parameter values, the larger number of clones was not favored based on the silhouette score. This indicates that the additional clones exhibited low within-clone similarity.

Breast cancer sample:

Prostate cancer sample:

In our opinion, the silhouette score, as suggested by the Authors of STARCH, is a good strategy for choosing the number of clusters when prior knowledge is absent. The silhouette score increases with the similarity within clusters as opposed to the similarity between clusters. We also want to emphasize that we lack prior knowledge of the number of copy number clones, which cannot be derived from WES. Our analysis using Canopy incorporates copy number

information inferred using FalconX, but this information is computed for bulk tissue and identifies clones based on single-point mutations, not copy number changes. We are not aware of any tool capable of inferring copy number clones from WES.

While I appreciate the addition of the scRNA-seq data to validate the clusters – additional methods are required to evaluate how the authors funneled their cell count down to ~300 cells for analysis. Based on how much tumor content is present, I am surprised by the total tumor cells present in this analysis.

Thank you for your comment. The initial cell count of approximately 300 cells originated from the use of a 384-well plate for the single-cell RNA sequencing (scRNA-seq) experiment. The data was filtered based on the following criteria: mitochondrial expression fraction < 0.1 , gene expression sum $> 10,000$, and the number of captured genes $> 3,000$. In this revision, an additional filtering step was applied to retain only the cells collected from the spatially transcriptomic (ST) sections utilized in the study, resulting in a final dataset of 120 high-quality cells. Subsequently, the analysis was performed using the top 100 variable genes. These details are now clearly explained in the **Methods** section on **page 23**, under the "**ScRNA-seq experiments for breast cancer samples**" section.

Please note that the number of cells analyzed in this revised version of the manuscript is smaller than in the previous one. This is because we have repeated the analysis, retaining only the cells that were sampled from the same regions as the sections included in our study.

Further – could you also utilize mutation mapping on the single cell data to further validate your claims of the clusters for overlapping samples. I have found the current data presented in Extended Data Fig 21 to be quite confusing but think this could be a very strong finish to your current publication if corrected appropriately. Are all tumor cells from all three breast cancer samples shown in this figure? Is each point a cell?

Thank you for your feedback. Following your suggestion, we revised the **Extended Data Fig 21** caption to make it clearer. Each point is a cell and each cross is a clone expression profile inferred by Tumoroscope. The cells are from the three breast samples.

Moreover, following your suggestion, we examined the mutations identified in the scRNA-seq data and obtained the read counts for each mutation. Then, we compared these mutations with those identified in WES and spatial transcriptomics (ST). Below, SB1, SB2, and SB3 represent mutations found in both single-cell and WES data for each of the three sections that we analyzed, with a p-value < 0.1 . Additionally, "Tree mutations" refer to those used in constructing the phylogenetic tree, meaning mutations identified in both ST and WES.

Unfortunately, only 7 mutations from the single-cell data across the three sections were also present in our phylogenetic tree. This may suggest that the spatial transcriptomics (ST) samples were collected from regions that were distant enough from those where the single cells were sampled to result in a small overlap in mutations between the ST and scRNA-seq data. Moreover, the overall number of mutations detected in single cells was low and weakly supported. This suggests that the sequencing coverage for scRNA-seq, while sufficient for characterizing gene expression, was inadequate for reliably calling mutations. Consequently, these limitations prevented us from reliably comparing the clones identified in ST and WES data with the mutations identified in scRNA-seq. Therefore, in the manuscript, we rely on validating the inferred expression profiles of the clones based on ST data with the expression levels measured in single cells.

Minor Comments:

Thank you for providing the higher resolution H&E images. I still think it is worth providing H&E images that are unmarked adjacent to the Annotated H&E images.

As indicated at the end of the **Data Availability** section, all H&E images, so also the unmarked, adjacent ones, were deposited and are available at zenodo <http://doi.org/10.5281/zenodo.10255434>. We only show one of each adjacent images in Figure 4 because they are almost identical to each other.

Line 68-70 - This sentence in introduction should be supported by a literature reference: "For some cancers, only very few or no copy number alterations occur during disease progression."

Thank you for this comment! We added the reference.

Line 104 - This statement seems redundant since alternated reads are reads containing mutations? If this is not the case please clarify. "alternated reads and the total number of reads

for each mutation (mutation coverage)". Maybe it should read as what is indicated in Line 110 instead?

We removed "for each mutation" in line 104.

Y-axis label for Figure 4c is missing.

It is fixed now.

Fig 4b – legend of color bar is not annotated.

Thank you for this comment. It is fixed now. The legend is annotated in the figure and explained in figure caption.

Fig 4g – colored spots in the cardelino figure are not annotated.

It is fixed now.

Figure 3 caption – presentation is misspelled.

It is fixed now.

Extended Data Figure 6 – Having a few more values on the y axis will be helpful for interpretation. Further if the spots could include common variant/other variant annotation.

We increased the number of values in both x and y axis. We showed all the shared variants in this figure.

We utilized nucleotide information from the WES data to select those alterations for which the specific alternative nucleotide matched consistently between the WES and ST data. We have included this information in Methods; **page 25**, paragraph 1.

In extended data fig 9 – an annotation is needed for the color legend. Is it showing the number of reads that are alternated/total etc? Why is the read count less than 1? Or is it the proportion of reads?

The legend is added now. It shows the number of alternated or total reads per cell and therefore it is divided by the number of cells.

COSMIC should be capitalized in Fig. 3 caption.

It is fixed now.

I appreciate the addition of Extended Data Figures 8-9. For extended data fig 9 please modify the color scheme as it is very hard to interpret the read coverage given the outliers really take over the entire plot.

We have inspected the count distributions and corrected **Extended data Figure 9** panels **c** and **d**, showing the values after the removal of outliers, which indeed have previously skewed the presentation.

I appreciate the time put into making the code accessible and available to support the work presented in this study.

We would like to add that the manuscript went through a language check, which resulted in several small edits to the text, which were also highlighted in blue. We also went through all figures in the manuscript and added missing ticks, labels, and missing color legends wherever it was necessary, on top of what was spotted by the Reviewer. Finally, the last paragraph of the Introduction of the previous version of the manuscript was integrated into the last paragraph of the Discussion, where it fits better.

Reviewer #2 (Remarks on code availability):

Ideally I would have relevant data to test through tumoroscope to make sure the functionality all makes sense and usable but do not currently have the bandwidth with the current submission deadlines.

If the authors address some of the remaining questions/comments I feel this work is acceptable for publication and will be useful for the community.